# A R-loop sensing pathway mediates the relocation of transcribed genes to nuclear pore complexes

Arianna Penzo [1], Marion Dubarry [2,6], Clémentine Brocas[3], Myriam Zheng [1], Raphaël M. Mangione [1], Mathieu Rougemaille [4], Coralie Goncalves[1], Ophélie Lautier[1], Domenico Libri[5], Marie-Noëlle Simon [2], Vincent Géli [2], Karine Dubrana [3] & Benoit Palancade [1] ✉

Nuclear pore complexes (NPCs) have increasingly recognized interactions with the genome, as exemplified in yeast, where they bind transcribed or damaged chromatin. By combining genome-wide approaches with live imaging of model loci, we uncover a correlation between NPC association and the accumulation of R-loops, which are genotoxic structures formed through hybridization of nascent RNAs with their DNA templates. Manipulating hybrid formation demonstrates that R-loop accumulation per se, rather than transcription or R-loop-dependent damages, is the primary trigger for relocation to NPCs. Mechanistically, R-loop-dependent repositioning involves their recognition by the ssDNA-binding protein RPA, and SUMO-dependent interactions with NPC-associated factors. Preventing R-loop-dependent relocation leads to lethality in hybrid-accumulating conditions, while NPC tethering of a model hybrid-prone locus attenuates R-loop-dependent genetic instability. Remarkably, this relocation pathway involves molecular factors similar to those required for the association of stalled replication forks with NPCs, supporting the existence of convergent mechanisms for sensing transcriptional and genotoxic stresses.

The three-dimensional organization of the nucleus plays a central role in the regulation of several genomic transactions, including transcription and DNA repair, thus contributing to the maintenance of genome homeostasis. Among the structural components of the nucleus that shape its compartmentalization are nuclear pore complexes (NPCs), which are conserved, megadalton-sized multiprotein assemblies embedded within the nuclear envelope and built from multiple copies of ~30 subunits called nucleoporins (Nups)[1]. While scaffold Nups sub-complexes delineate a central channel in which nucleo-cytoplasmic exchanges occur, peripheral components, i.e., the cytoplasmic filaments and the nuclear basket, extend contacts towards the cytoplasm and the nucleoplasm. In this way, nuclear pore complexes notably establish interactions with specific regions of the genome, beyond their canonical role in the selective transport of proteins and RNAs[2]. This is well exemplified in budding yeast, where several inducible loci, including galactose-activated and heat shock genes,

[1]Université Paris Cité, CNRS, Institut Jacques Monod, F-75013 Paris, France. [2]Marseille Cancer Research Center (CRCM), U1068, Institut National de la Santé et de la Recherche Médicale (INSERM), UMR7258, Centre National de la Recherche Scientifique (CNRS), Aix Marseille University, Institut Paoli-Calmettes, Equipe Labélisée Ligue, 13273 Marseille, France. [3]Université Paris Cité, Université Paris-Saclay, INSERM, iRCM/IBFJ CEA, UMR Stabilité Génétique Cellules Souches et Radiations, Fontenay-aux-Roses, France. [4]Université Paris-Saclay, CEA, CNRS, Institute for Integrative Biology of the Cell (I2BC), 91198 Gif-sur-Yvette, France. [5]Institut de Génétique Moléculaire de Montpellier, Univ Montpellier, CNRS, Montpellier, France. [6]Present address: Univ Lyon, Université Claude Bernard Lyon 1, INSA-Lyon, CNRS, UMR5240, Microbiologie, Adaptation et Pathogénie, F-69622 Villeurbanne, France. ✉e-mail: benoit.palancade@ijm.fr

relocate to the nuclear periphery and associate with NPCs upon transcriptional activation[3-13]. While proximity to the nuclear pore complex may couple transcription with mRNA processing and export, thus positively impacting gene expression[3,10,14], the physiological significance of gene repositioning, a.k.a. "gene gating"[15], still remains debated. Strikingly, DNA lesions, e.g., unrepairable DNA double-strand breaks (DSBs)[16], eroded telomeres[17], or challenged replication forks[18-21], also cause the relocation of genomic regions to nuclear pore complexes in yeast cells. In these situations, NPC association has been shown to locally impact DNA repair pathway choices, thus contributing to the maintenance of genetic integrity[22]. Remarkably, nucleoporins similarly interact with transcribed or damaged loci in several distant species, in which their loss-of-function can trigger changes in gene expression or DNA damage, pointing to the functional importance of NPC-chromatin interactions[22-24].

Repositioning typically involves diffusive or active motion of chromatin domains within the nucleus[23,24]. Anchoring of specific genomic regions to NPCs is further achieved through protein-protein contacts involving DNA- and NPC-bound factors, for instance the mediator and TREX-2 complexes, whose association bridges the promoter of activated *GAL* genes with the nuclear basket[7,25]. The interactions between chromatin and NPCs also involve SUMOylation, a post-translational modification that relies on the covalent addition of the SUMO (small ubiquitin-like modifier) polypeptide to its protein targets. SUMO moieties are covalently coupled to lysine residues through an enzymatic cascade involving an E1 activating enzyme, an E2 conjugating enzyme and an E3 ligase; removed through the action of SUMO-proteases; and recognized by SIM (SUMO interaction-motifs)-containing proteins, including SUMO-targeted ubiquitin ligases (STUbLs), which can target modified substrates to proteasomal degradation[26,27]. Remarkably, the SUMO pathway is itself compartmentalized within the yeast interphasic nucleus, with the three SUMO ligases (Siz1, Siz2, Mms21) being localized in the nucleoplasm and/or at the inner nuclear membrane[28,29], while the essential SUMO-protease Ulp1 and the Slx5/Slx8 STUbL are mostly restricted to NPCs[16,30]. In this respect, highly expressed or inducible genes harbor high levels of SUMOylation[31], and their repositioning to NPCs requires both the SUMO E3 ligase Siz2 and the SUMO-protease Ulp1[14,29,32]. Similarly, SUMOylation waves occur at DNA lesions[33,34], and the relocation of damaged chromatin to NPCs involves SUMO ligases and SIM-containing NPC-associated factors[19,20,35-37]. Beyond these common signals, it remains however to be understood whether gene gating and damage relocation utilize redundant or overlapping pathways.

Another process connecting high transcriptional activity to genetic instability is the formation of R-loops, which are three-stranded structures formed through the annealing of nascent RNAs onto their DNA templates, thus displacing single-stranded DNA moieties. In yeast, R-loops preferentially form at highly expressed loci[38,39] and their unscheduled accumulation ultimately leads to replication stress and DSBs accumulation[40]. While the formation of R-loops is sterically prevented by the coating of the transcripts with RNA-binding factors, such as the THO complex[41] and the spliceosome[42], their removal from the genome involves dedicated enzymes, including ribonucleases of the RNase H family and DNA:RNA helicases[43]. How R-loops are detected and handled within the nuclear environment is however poorly understood. Notably, coating by the ssDNA-binding complex RPA (replication protein A) has been proposed to sense R-loops and possibly promote their removal through recruitment of RNase H1 in human cells[44]. Strikingly, R-loop accumulation and increased R-loop-dependent genetic instability were scored in yeast nucleoporin mutants[45,46], raising the possibility that the NPC could also function in hybrid metabolism. In this line, artificial tethering of an R-loop forming gene to the nuclear pore complex was reported to attenuate hybrid levels[46]. Altogether, these findings prompted us to explore whether R-loops themselves could act as a signal for repositioning to NPCs. By combining biochemical and live imaging approaches, we further examined the signals and pathways potentially mediating the association of R-loops with nuclear pore complexes, in light of our knowledge of chromatin-NPC interactions. Finally, we investigated the functional impact that proximity to the pore could exert on R-loop metabolism.

## Results

### Genome-wide association between R-loop-prone loci and NPCs

To investigate the relationships between R-loop formation and gene positioning, we compared the localization of genomic NPC contact sites to maps of DNA:RNA hybrid distribution that we or others had previously generated[38,39]. For this purpose, we performed chromatin immunoprecipitation coupled to sequencing (ChIP-seq) using a functional, myc-tagged version of the scaffold nucleoporin Nic96 as a bait (Fig. 1a). Several NPC contact sites were observed within protein-coding genes, which were more highly transcribed, on average, than the rest of the genome (Supplementary Fig. 1a). Since DNA:RNA hybrid formation also correlates with transcription[38,39] (Supplementary Fig. 1b), we restricted our analysis to the most highly expressed genes, which were further categorized according to their intron content, a *cis*-acting modulator of R-loop formation[42]. In this way, we were able to compare NPC association between two equally sized groups of genes with similar transcription rates, transcript lengths and base contents (Supplementary Fig. 1c, Supplementary Table 1), but distinctive R-loop levels (Supplementary Fig. 1d, e). Strikingly, R-loop-prone, intronless loci displayed higher Nic96 occupancy over their gene bodies as compared to their R-loop-depleted, intron-containing counterparts (Fig. 1b, c). Furthermore, the extent of Nic96 enrichment over intronless loci correlated with their propensity to form R-loops (Fig. 1b, c, top panels, genes ranked by R-loop levels; Fig. 1d). To confirm this finding, we performed the same analysis on independent datasets of NPC-bound genes, which were previously obtained using distinct scaffold nucleoporins as baits in ChIP-on-chip experiments[47] (Fig. 1e, f). Similarly, highly expressed, R-loop-forming intronless genes showed enhanced association with Nup170 and Nup157 compared to the intron-containing group (Fig. 1e, f). Importantly, enhanced association to NPCs was observed for a large fraction of the intronless gene set, as notably pointed out by Nup157 ChIP analyses (Fig. 1f). The fact that NPC association is less pronounced for intron-containing loci, despite being similarly transcribed as the intronless group, further rules out that the detected signals reflect the intrinsic bias of ChIP experiments for highly transcribed regions[48]. Overall, our genome-wide analyses indicate that the propensity of genes to form R-loops correlates with their association with NPCs.

### A reporter assay to probe R-loop-dependent relocation to NPCs

To directly assess whether hybrid formation triggers gene localization at NPCs, we engineered a reporter locus in which R-loop accumulation can be locally modulated, and further tracked its nuclear position through live imaging. Since we previously reported that high levels of transcription trigger R-loop formation on the GC-rich *YAT1* ORF, both in vitro and in vivo[42], we chose to insert this bona fide R-loop forming sequence within the chromosome II *GAL* locus under the control of the inducible *GAL1-10* promoter (Fig. 1g). To further enhance hybrid accumulation, we used a mutant of the THO complex (*mft1Δ*, hereafter labeled *tho*), which triggers R-loops and R-loop-dependent genetic instability on the *YAT1* gene[42,49]. Conversely, to locally attenuate hybrid formation, we inserted at the 5' of the *YAT1* transgene a short artificial intron, which alleviates R-loop formation and R-loop-associated genotoxicity in *cis* without decreasing mRNA levels[42] (Supplementary Fig. 1f). In these different strains, direct repeats flanking the *YAT1* reporter permitted the quantification of R-loop-dependent recombination events, which reconstitute a functional *LEU2* prototrophy marker (Fig. 1g). Importantly, this assay confirmed that the integrated

reporter exhibits increased R-loop-dependent genetic instability in *tho* mutants, a phenotype rescued by the insertion of the intron (Supplementary Fig. 1g), in agreement with our previous observations using plasmid-borne versions of the same constructs[42]. An array of LacI-GFP-bound tandem repeats of the bacterial Lac operator inserted at the same locus allowed us to visualize the position of the reporter gene with respect to NPCs, which were detected using a mCherry-tagged version of the Nup49 nucleoporin (Fig. 1h; Supplementary Fig. 1h). We assigned peripheral localization to the loci positioned in the most external of the three equivolumetric zones in which the nucleus is segmented for image analysis (Fig. 1h). In this assay, the position of a locus can range from ~33% in zone 1 if randomly localized, to 60–70% for well-characterized loci dynamically associated to the NE (*e.g.* telomeres[50]). Of note, we only analyzed unbudded, G1 cells to avoid the

possibly confounding effects of replication-dependent relocation to NPCs[22]. As expected, in the absence of transcription (glucose-containing medium), the reporter gene appeared randomly distributed in the nucleus in both wild-type and *tho* mutant cells, regardless of its intron content (Fig. 1i). However, upon transcriptional activation (galactose-containing medium), the fraction of cells with intronless *YAT1* located at the nuclear periphery significantly increased in wild-type cells, a phenotype further enhanced in the R-loop-accumulating *tho* mutant (Fig. 1i). Strikingly, the presence of the intron completely abolished *YAT1* relocation to the nuclear envelope in both wild-type and *tho* mutant cells (Fig. 1i). Altogether, these results support the idea that R-loop accumulation triggers relocation of an inducible locus to NPCs, mirroring the genome-wide observations reported above for constitutively expressed genes.

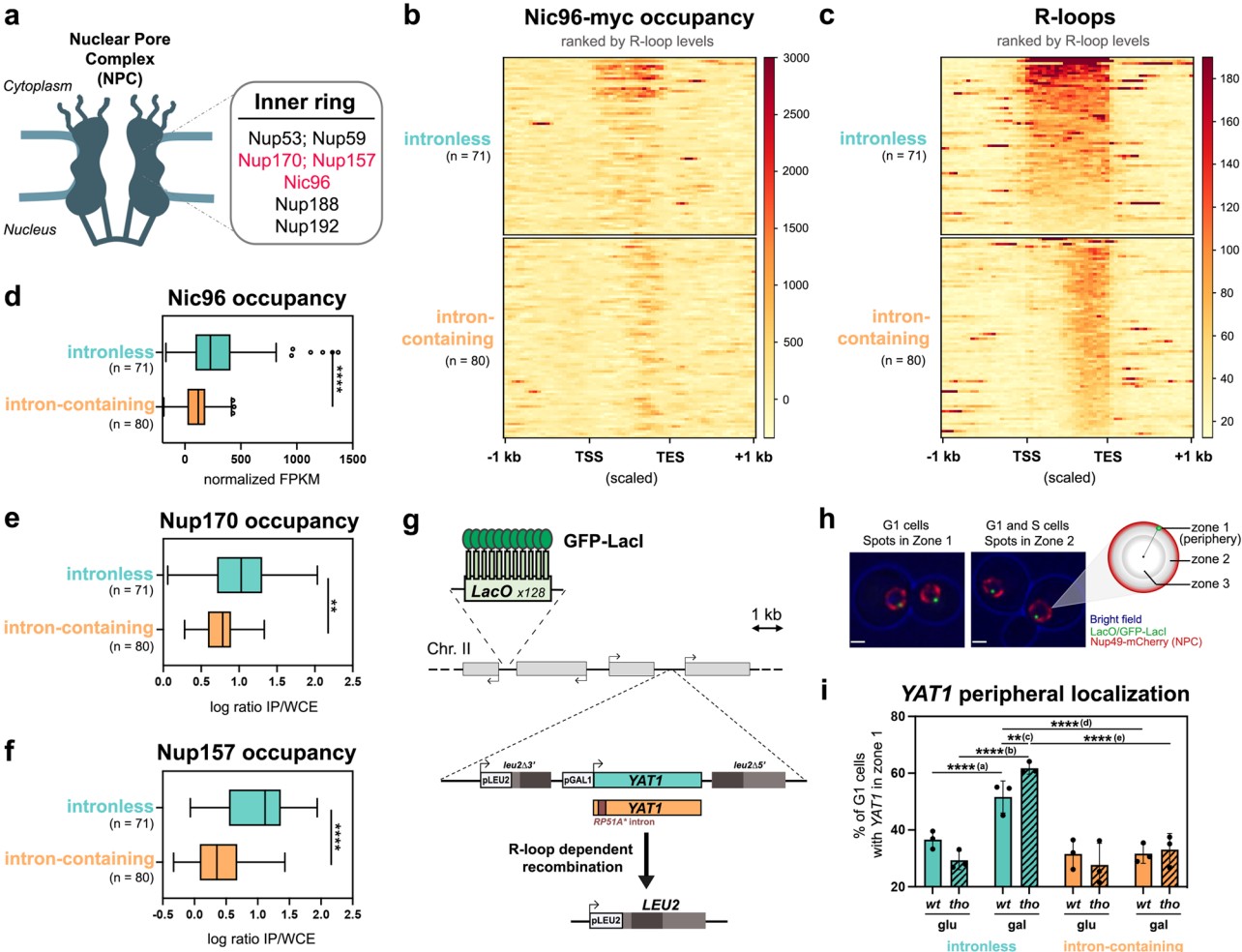

**Fig. 1 | R-loops can be a signal for repositioning to NPCs. a** Schematic representation of the yeast nuclear pore complex (NPC). The inner ring nucleoporins highlighted in red were used as baits in ChIP-seq (Nic96, this study) and ChIP-on-chip (Nup170, Nup157)[47]. **b** Heatmap analysis of Nic96 occupancy at highly transcribed intronless and intron-containing genes, aligned at their Transcription Start Site (TSS) and Transcription End Site (TES). Only the regions between the TSS and the TES are scaled. Genes are grouped based on their intron content and ranked according to their R-loop levels (**c**)[38]. **c** Heatmap analysis of R-loop levels[38] at highly transcribed intronless and intron-containing genes, aligned as in (**b**) and ranked according to R-loop signals. **d**–**f** Average Nic96, Nup170 or Nup157 occupancies at highly transcribed intronless and intron-containing genes (Nic96: normalized FPKM; Nup170 and Nup157: average log2 [IP/whole cell extract] for all the probes covering a given genomic feature). Boxes extend from the 25th to 75th percentiles, with the median displayed as a line. The whiskers mark the minimum and maximum, displaying as individual points the values that fall outside of 1.5 time the

inter-quartile range of the first or third quartile (Tukey's definition). Statistical test: two-sided Mann-Whitney-Wilcoxon test; **d**, ****$p < 10^{-4}$; **e**, **$p = 1.6 \times 10^{-3}$; **f**, ****$p < 10^{-4}$. **g** Principle of the *YAT1* integrated reporter construct. **h** Principle of the zoning assay. The locus of interest appears as a bright green dot, relative to the nuclear envelope, stained by the Nup49-mCherry nucleoporin. The nucleus is divided in three equivolumetric regions and the dots localizing at the outermost region are scored as peripheral (zone 1). Scale bar = 1 μm. **i** Fraction of G1 cells (%) showing intronless or intron-containing *YAT1* in zone 1 (mean ± SD, *n* = 3 independent experiments), in *wt* or *tho* (*mft1Δ*) mutant cells grown in glycerol-lactate medium and further treated with either glucose (glu) or galactose (gal) for 5 h. Statistical test: two-sided Fisher's exact test; *P*-values were calculated based on the total number of counted cells (between 294 and 480 cells/condition); (a), $p = 9.7 \times 10^{-5}$; (b), $p < 10^{-8}$; (c), $p = 2.56 \times 10^{-3}$; (d), $p = 1.41 \times 10^{-8}$; (e), $p < 10^{-8}$. Source data are provided as a Source Data file.

## Stress-induced transcriptional activation leads to R-loop dependent relocation to NPCs

To expand our findings, we wondered whether stress situations involving the coordinated transcriptional induction of multiple responsive loci would similarly result in their R-loop-dependent repositioning to NPCs. To achieve this, we focused our attention on the heat shock (HS) response since it induces high levels of transcription at heat shock genes (*e.g. HSP104*), some of which were previously reported to relocate to NPCs upon activation[8,51]. Intriguingly, it had been reported that the association of HS-activated loci with NPCs is enhanced in cells lacking the THO complex[52,53].

First, we investigated whether HS-dependent transcription triggers R-loop formation at responsive loci, using two distinct yet complementary strategies: (i) DNA:RNA hybrid immunoprecipitation (DRIP) with the hybrid-specific S9.6 monoclonal antibody (Supplementary Fig. 2a, left panel), which typically reveals RNase H-sensitive signals on gene bodies, and (ii) RNAse H ChIP (R-ChIP; Fig. 2a, top panel), which reportedly outperforms DRIP in detecting native, transient hybrids[54]. Strikingly, DRIP failed to detect HS-dependent hybrids at the *HSP104* locus, while it successfully identified R-loop formation at a constitutively transcribed locus (*YEF3*, Supplementary Fig. 2a, right panel). In contrast, R-ChIP revealed the specific enrichment of a tagged version of yeast RNase H1 (scRNH1) at *HSP104* upon heat shock, indicating the occurrence of transient, rather unstable hybrids forming at this locus upon transcriptional activation (Fig. 2a, bottom panel).

To further evaluate the role of these hybrids in the interaction of HS genes with NPCs, we interfered with their formation by combining the R-loop-accumulating *tho* mutant with the overexpression of human RNH1 (hsRNH1), a classical strategy to probe R-loop-dependent phenotypes[42,55]. We also took advantage of previous observations reporting that the enhanced peripheral localization of HS genes in *tho* mutants is reflected by their biochemical co-fractionation with NPCs in heavy chromatin (HC) isolates[52] (Fig. 2b). Strikingly, the increased occurrence of *HSP104* DNA in heavy chromatin fractions after heat shock was suppressed by hsRNH1 overexpression in *tho* mutant cells (Fig. 2c). Of note, co-fractionation was also reduced, albeit to a lower extent, upon over-expression of the tagged version of yeast RNH1 used in R-ChIP experiments (Supplementary Fig. 2b). Importantly, microscopy analyses of *HSP104* locus position using strains with LacO arrays inserted downstream of the gene[52] revealed that its HS-induced relocation to the nuclear periphery was enhanced in the R-loop-accumulating *tho* mutant (Supplementary Fig. 2c, d), mirroring the results from our fractionation analyses (Fig. 2c). Of note, the repositioning at the nuclear periphery probed by our imaging assays likely reflects gene association with NPCs rather than with other NE domains, as supported by the colocalization of the *HSP104* locus with the typical NPC clusters observed in the *nup133Δ* nucleoporin mutant (Supplementary Fig. 2e, f). Furthermore, this R-loop-dependent association with NPCs was not detected for housekeeping genes (*YEF3, ACT1*) or an intergenic region, but was similarly scored in *tho* mutant cells for other HS-induced loci (*e.g. GSY2, PAU17, SSA4*) or the *YAT1* R-loop-forming reporter, supporting its dependence on high transcription and R-loop accumulation (Supplementary Fig. 2g, h).

To determine whether other genes of the HS regulon similarly display R-loop-induced repositioning, we further mapped HS-induced genomic NPC contact sites by ChIP-seq, using the same nucleoporin bait as above (Nic96). HS triggered the appearance of extended regions of contact between the NPC and gene bodies in wild-type cells, with increased Nic96 enrichment in the *tho* mutant, as exemplified in Fig. 2d. When specific NPC-associated peaks were ranked by size, this increase was deeply marked for long regions (>1 kb; Fig. 2e). Gene ontology analysis of the genes displaying such extended contacts finally revealed an over-representation of heat shock responsive loci (GO: "protein folding", $p = 1.5e{-}4$; "response to temperature stimulus", $p = 1.25e{-}3$). Finally, Nic96 ChIP assays performed in wild-type or *tho*

mutant cells overexpressing hsRNH1 further confirmed the association of NPCs with several of these HS-induced loci (*HSP104, GSY2, PAU17, SSA4*) and its dependence on R-loop accumulation (Supplementary Fig. 2i). Overall, these data indicate that R-loop formation can also act as a trigger for gene relocation to NPCs in the case of a coordinated transcriptional response impacting several distant loci.

## R-loop-dependent gating defines an original NPC relocation pathway

We next investigated the relationships between this newly uncovered R-loop-dependent gene repositioning process and other situations where specific chromatin regions also interact with NPCs (Fig. 3a). Highly expressed and inducible genes were previously reported to associate with nuclear pore complexes during the course of transcriptional activation, in a gene gating pathway requiring the NPC-bound TREX-2 (Transcription and Export) complex[11,13,25]. However, *tho* mutants, in which we scored increased association of R-loop-forming loci with NPCs, have globally reduced transcription rates[41,56], as notably reported for the *YAT1* gene[42,49]. Moreover, TREX-2 mutants (e.g. *sac3Δ*) trigger association of the *HSP104* locus with NPCs at 37 °C, as revealed by chromatin fractionation (Fig. 3b). It is likely that R-loop formation also acts as a signal for relocation to NPCs in this mutant, given the reported role of TREX-2 in preventing RNA-dependent genetic instability[57], including at the *YAT1* gene[42]. The fact that R-loop-dependent-repositioning still occurs in the absence of TREX-2 further supports the notion that this relocation pathway is genetically distinguishable from canonical gene gating.

We also considered whether R-loop-dependent DNA breakage or replication impairment might be the actual trigger for NPC repositioning, as both DSBs and blocked replication forks were previously shown to relocate to nuclear pore complexes[22] (Fig. 3a). To investigate the involvement of R-loop processing into DSBs in relocation, we combined the *tho* R-loop-accumulating mutant with the inactivation of enzymes described to trigger R-loop cleavage in yeast (Fig. 3a), i.e., the nucleotide excision repair factor Rad2[58], the DNA mismatch repair protein Mlh3 and the cytosine deaminase Fcy1[59]. None of the analyzed double mutants showed decreased *HSP104*-NPC co-fractionation as compared to the single *tho* mutant (Fig. 3c), indicating that R-loop-dependent damage is not responsible of *HSP104* peripheral localization upon HS. Noteworthy, the genotoxicity of R-loops mainly arises from their encounter with the replication machinery[60] (Fig. 3a). To assess whether R-loop-induced repositioning could stem from interferences with replication, we repeated the chromatin fractionation assay in cells synchronized in G1 by alpha-factor treatment. In these conditions, *tho* mutant cells still displayed an increased occurrence of the *HSP104* gene in the NPC fraction, similar to asynchronous cultures (Fig. 3d), in agreement with our microscopy observations in G1 cells (Supplementary Fig. 2d). Altogether, these observations establish that R-loop-dependent loci relocation to the periphery occurs independently of DNA damage and replication.

## R-loop-dependent relocation to the nuclear pore complex requires ssDNA coating by RPA

Our data define an original pathway for NPC relocation, hereafter referred to as "R-loop gating", in which R-loop accumulation, rather than increased transcription, damage formation or interference with replication, is the primary cause of repositioning. Among the distinctive structural features of R-loops that could be recognized prior to relocation are the ssDNA moieties of these three-stranded structures. We therefore directed our attention to RPA, the main cellular ssDNA-binding complex, which was previously localized to transcribed genes in yeast and associated with R-loop sensing and resolution in mammalian cells[44,61]. To specifically assess the presence of RPA at R-loop forming genes without the confounding effect of its replication-dependent recruitment, we used a strand-specific RPA ChIP-seq

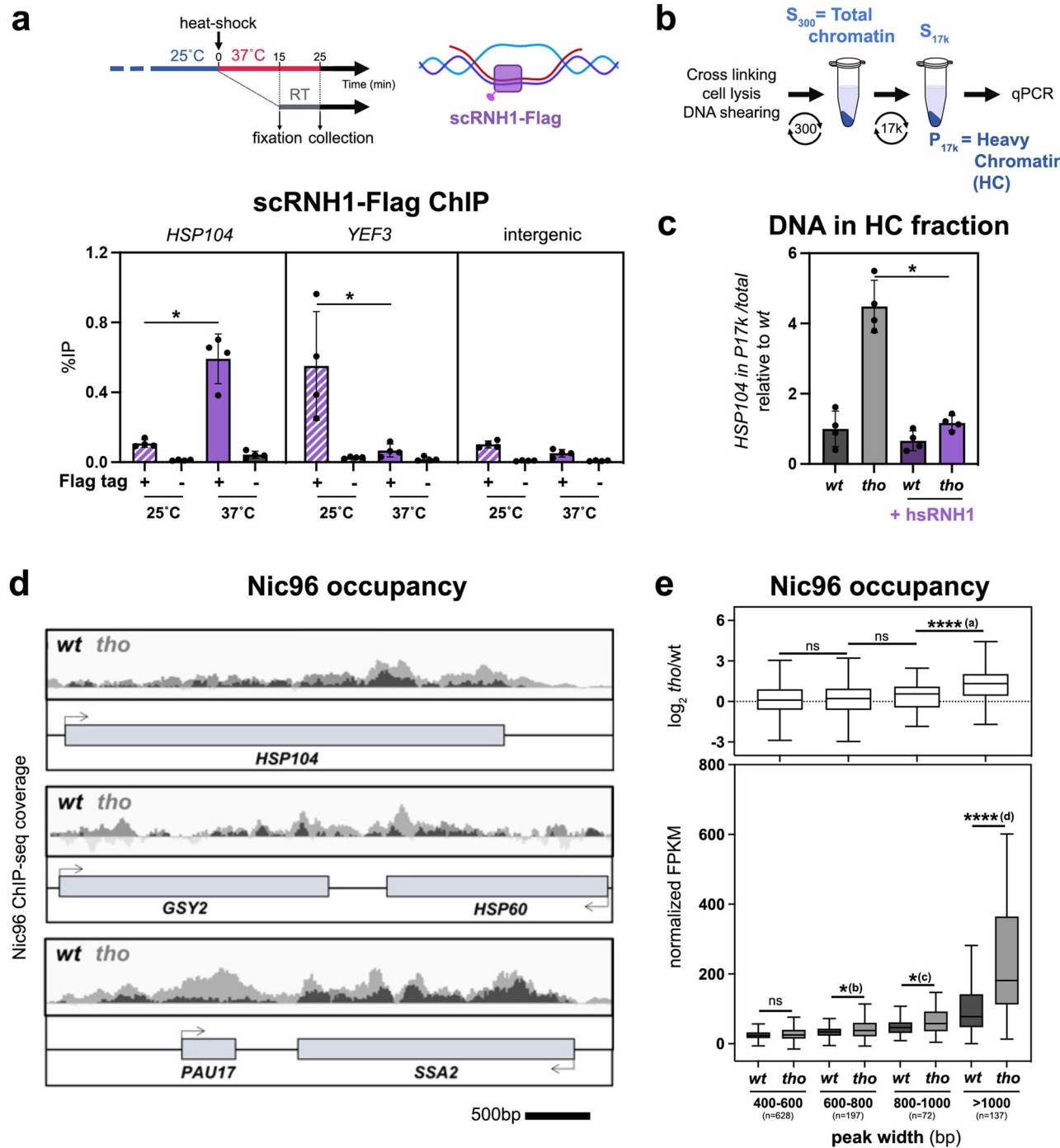

**Fig. 2 | Stress-induced transcriptional activation leads to R-loop dependent relocation to the NPC. a** Top: timeline of the heat shock and fixation procedure and schematic representation of scRNase H1 binding to the DNA:RNA hybrid within an R-loop; Bottom: scRNase H1-Flag occupancy was analyzed at the indicated loci by ChIP-qPCR in *wt* cells transformed with either an empty vector, or the *tetOFF-scRNH1-Flag* construct, and further grown at 25 °C or heat shocked at 37 °C for 15 min (% of immunoprecipitation; mean ± SD, *n* = 4 independent experiments). Statistical test: two-sided Mann-Whitney-Wilcoxon test; *$p = 2.86 \times 10^{-2}$. **b** Principle of the differential chromatin fractionation procedure. The presence of the gene of interest in the pellet (P17k) and supernatant (S17k) fractions is evaluated by qPCR. **c** qPCR-based quantification of the amount of DNA from the *HSP104* locus in heavy chromatin (HC) fractions from *wt* or *tho* (*mft1Δ*) mutant cells transformed with either an empty vector or the *GPD-hsRNH1* construct (+hsRNH1), and heat shocked

at 37 °C for 15 min (% of *HSP104* in P17K relative to total [S17K + P17K]; mean ± SD, *n* = 4 independent experiments). Statistical test: two-sided Mann-Whitney-Wilcoxon test; *$p = 2.86 \times 10^{-2}$. **d** Integrative Genomics Viewer (IGV) representative screenshots of Nic96 ChIP-seq coverage in *wt* or *tho* (*mft1Δ*) mutant cells heat shocked at 37 °C for 15 min. Scale bar, 500 bp. **e** Nic96 enrichment (bottom panel, normalized FPKM; top panel, log2 [*tho/wt*]) in *wt* or *tho* mutant cells heat shocked at 37 °C for 15 min. Nic96-bound regions identified through peak calling were categorized according to peak width (bp). The number of regions is indicated for each category. Box-plots are defined as above (Fig. 1d–f). Outliers identified according to Tukey's definition are not represented on this scale but have been included in statistical analyses. Statistical test: two-sided Mann-Whitney-Wilcoxon test; (a), $p < 10^{-4}$; (b), $p = 3.59 \times 10^{-2}$; (c), $p = 1.58 \times 10^{-2}$; (d), $p < 10^{-4}$. Source data are provided as a Source Data file.

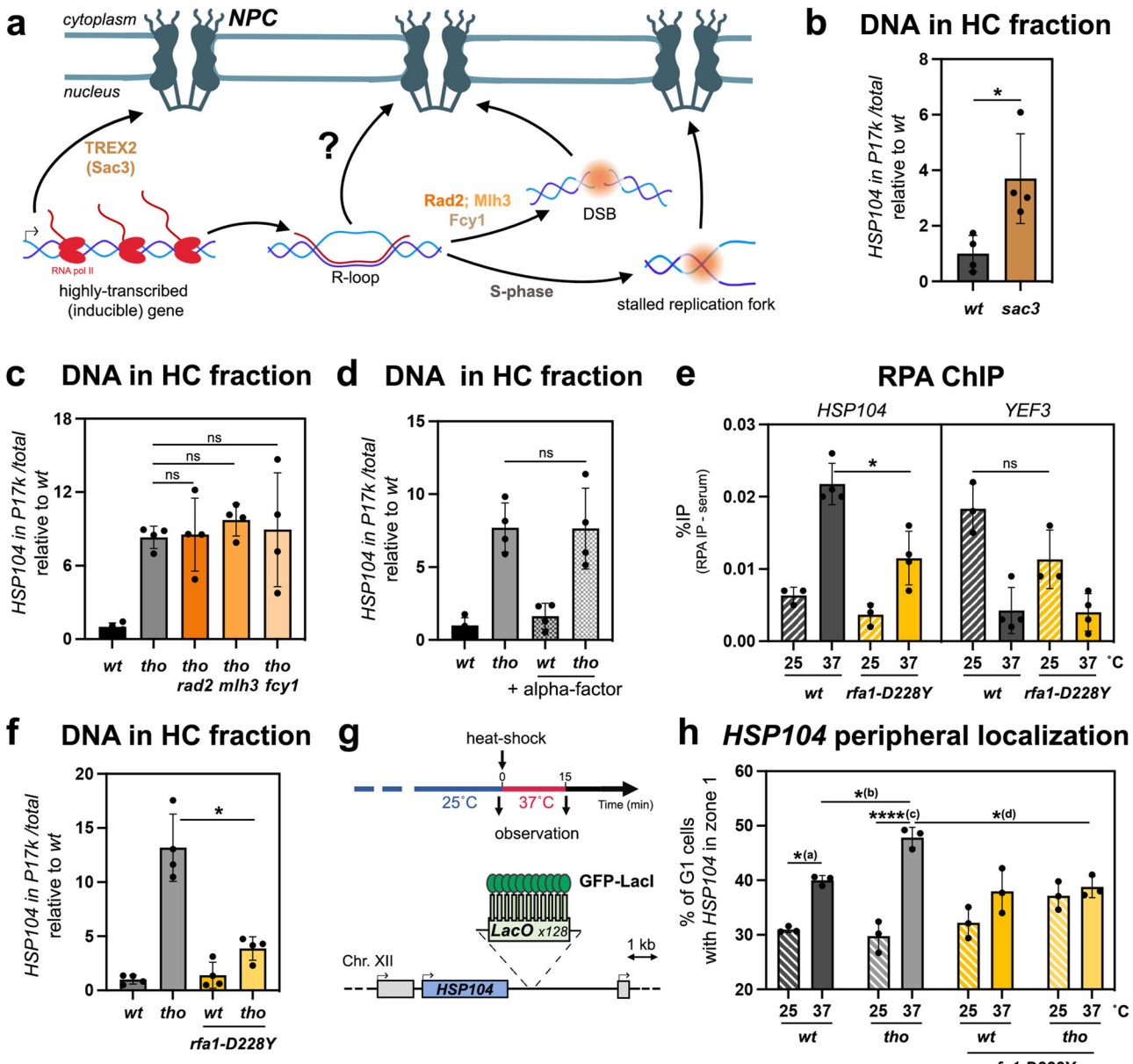

**Fig. 3 | R-loop-dependent relocation to NPCs requires ssDNA coating by RPA.**
**a** Schematic representation of characterized pathways mediating the relocation of highly transcribed genes and damaged loci to the nuclear periphery, and of their possible relationship with R-loop gating. **b**, **c** qPCR-based quantification of the amount of DNA from the *HSP104* locus in heavy chromatin (HC) fractions from the indicated strains heat shocked at 37 °C for 15 min (% of *HSP104* in P17K relative to total [S17K + P17K]; mean ± SD, *n* = 4 independent experiments, relative to *wt*). Statistical test: two-sided Mann-Whitney-Wilcoxon test; *\*p* = 2.86 × 10⁻². **d** qPCR-based quantification of the amount of DNA from the *HSP104* locus in heavy chromatin (HC) fractions from from *wt* or *tho* (*mft1Δ*) mutant cells heat shocked at 37 °C for 15 min (% of *HSP104* in P17K relative to total [S17K + P17K]; mean ± SD, *n* = 4 independent experiments, relative to *wt*). When indicated, cells were arrested in G1 through alpha-factor treatment (+alpha-factor). Statistical test: two-sided Mann-Whitney-Wilcoxon test. **e** RPA occupancy was analyzed at the indicated loci by ChIP-qPCR in *wt* or *rfa1-D228Y* mutant cells, grown at 25 °C or heat shocked at 37 °C

for 15 min (% of immunoprecipitation; mean ± SD, *n* = 4 independent experiments). Statistical test: two-sided Mann-Whitney-Wilcoxon test; *\*p* = 2.86 × 10⁻². **f** qPCR-based quantification of the amount of DNA from the *HSP104* locus in heavy chromatin (HC) fractions from the indicated strains heat shocked at 37 °C for 15 min (% of *HSP104* in P17K relative to total [S17K + P17K]; mean ± SD, *n* = 4 independent experiments, relative to *wt*). Statistical test: two-sided Mann–Whitney–Wilcoxon test; *\*p* = 2.86 × 10⁻². **g** Schematic representation of the time-line of the procedure and of the tagged genomic *HSP104* locus used for microscopy experiments.
**h** Fraction of G1 cells (%) showing *HSP104* in zone 1 (mean ± SD, *n* = 3 independent experiments), in the indicated strains grown at 25 °C or heat shocked at 37 °C for 15 min. Statistical test: two-sided Fisher's exact test; *P*-values were calculated on the total number of counted cells (between 210 and 409 cells/condition); (a), *p* = 1.59 × 10⁻²; (b), *p* = 2.45 × 10⁻²; (c), *p* = 4.96 × 10⁻⁷; (d), *p* = 1.48 × 10⁻². Source data are provided as a Source Data file.

dataset obtained from non-cycling yeast cells[62]. By restricting our analysis of RPA occupancy to intronless and intron-containing highly transcribed genes (same groups as Fig. 1b, c), we scored RPA stretches along intronless gene bodies (Supplementary Fig. 3a, b), correlating with their R-loop levels (Fig. 1c, Supplementary Fig. 1d, e) and their association with NPCs (Fig. 1b, d–f). In contrast, RPA binding was nearly

undetectable on their R-loop-depleted, intron-containing counterparts (Supplementary Fig. 3a, b). Of note, RPA occupancy was similarly detected at forward and antisense strands in this ChIP assay (Supplementary Fig. 3a, b), an expected pattern since yeast R-loops are typically smaller (~150bps[63]) than sheared chromatin fragments (Supplementary Fig. 3c).

To confirm the correlation between RPA recruitment and R-loop gating, we used ChIP-qPCR to monitor RPA association to model loci in control and heat shocked cells. RPA recruitment was indeed observed onto *YEF3*, a member of the intronless gene group used above, in control conditions, yet was abolished upon HS (Fig. 3e, right panel), which reportedly represses its transcription[64,65]. In contrast, HS triggered RPA recruitment onto *HSP104* (Fig. 3e, left panel), concomitantly with the activation of this inducible locus. Importantly, RPA recruitment was reduced at both genes in a mutant impairing its association to ssDNA (*rfa1-D228Y*[66–68]), testifying the specificity of the detected signal (Fig. 3e).

To investigate whether RPA recruitment to R-loops is required for NPC repositioning, we assessed the effect of the *rfa1-D228Y* mutation on the relocation of the *HSP104* locus to the nuclear periphery upon inactivation of the THO complex. Strikingly, decreased RPA binding nearly abrogated the co-fractionation of *HSP104* with the nuclear pore complex (Fig. 3f), indicating the crucial role of RPA in NPC relocation. To confirm this finding, we proceeded to score the nuclear localization of the HS-induced *HSP104* locus by microscopy. Remarkably, impairing RPA association to ssDNA virtually suppressed the increase in *HSP104* peripheral localization in *tho* mutant cells (Fig. 3g, h; compare *tho* and *tho rfa1-D228Y*). These data support that RPA senses the formation of R-loops by coating their ssDNA moiety and mediates their relocation at nuclear pore complexes.

## SUMOylation events mediate R-loop-dependent NPC association

The establishment of contacts between transcribed chromatin and nuclear pore complexes was previously reported to involve random sub-diffusion of the targeted locus within the nucleus, followed by its capture at the nuclear periphery by virtue of gene-NPC interactions[69]. We thereby asked whether dedicated factors could mediate the interaction between nuclear pore complexes and RPA-bound R-loop-forming genes undergoing repositioning. Of note, RPA subunits were not previously identified in our proteomic analyses of nuclear pore complexes[70,71], suggesting the existence of indirect or labile interactions between this ssDNA-binding complex and NPCs. In light of the multiple reports indicating that SUMOylation, a highly reversible post-translational modification, can contribute to NPC-chromatin interactions[19,29,32,35] while targeting the RPA subunit Rfa1[20,36], we wondered whether this ssDNA-binding complex could be SUMOylated concomitantly with R-loop gating. To test this hypothesis, we expressed a polyhistidine-tagged version of SUMO (His-SMT3) under the control of its endogenous promoter, to avoid artefacts dues to overexpression, and purified SUMO-conjugates from yeast by denaturing affinity chromatography. Western-blot detection using Rfa1-specific antibodies did not reveal any modified bands in wild-type cells, yet uncovered a slower-migrating version of Rfa1 in a mutant of the NPC-associated SUMO deconjugating enzyme Ulp1 (*ulp1-333*), with a molecular weight compatible with mono-SUMOylation (Fig. 4a). Performing the same assay in a mutant strain expressing a non-SUMOylatable version of Rfa1, *rfa1-4KR*[72], further confirmed that this species corresponds to mono-SUMOylated Rfa1 (Fig. 4a). Strikingly, RPA SUMOylation increased upon heat shock induction (Fig. 4a; Supplementary Fig. 4a, b), suggesting that this modification occurs concomitantly with R-loop relocation.

To further characterize the involvement of SUMOylation in this process, we assessed the impact of the inactivation of several components of the SUMO pathway (Fig. 4b) on R-loop-NPC association using the same combination of biochemical and live imaging approaches as above. Remarkably, *HSP104* peripheral localization was completely suppressed upon removal of the SUMO-ligase domain of Mms21 (Fig. 4c, compare *tho* and *tho mms21-11*), a subunit of the cohesin-like Smc5/6 complex[73]. Similarly, *HSP104* co-fractionation with the nuclear pore complex was strongly alleviated in *mms21-11* mutant

cells (Fig. 4d), while it remained unperturbed upon the double inactivation of the two main yeast SUMO-ligases Siz1 and Siz2 (Supplementary Fig. 4c). Conversely, *HSP104* localization to NPCs was unchanged in a SUMO mutant unable to form poly-SUMO chains (*smt3-3KR*; Fig. 4e), suggesting that mono-SUMOylation events, as detected for Rfa1, are sufficient for repositioning. Consistently, preventing Rfa1 SUMOylation reduced the extent of *HSP104* peripheral localization (Fig. 4c, compare *tho* and *tho rfa1-4KR*) and co-fractionation with the nuclear pore complex (Fig. 4f). The fact that loss of Rfa1 SUMOylation (*rfa1-4KR*, Fig. 4f) does not fully phenocopy the RPA ssDNA-binding mutant (*rfa1-D228Y*, Fig. 3f) or the SUMO-ligase inactivation (*mms21-11*; Fig. 4d) suggests the existence of additional SUMOylation events, occurring downstream of RPA binding and involving Mms21 activity towards other factors, possibly associated with R-loops. While it was previously reported that Mms21 can SUMOylate RPA upon genotoxic stress[20], we could not assess whether this SUMO-ligase is also involved in the Rfa1 SUMOylation events detected upon HS in *ulp1* cells (Fig. 4a), because of the strict co-lethality between *MMS21* and *ULP1* inactivation[73].

Finally, to investigate the mechanisms by which R-loop-bound, SUMOylated RPA interacts with nuclear pore complexes, we assessed whether repositioning required SUMO-interaction motifs (SIM)-containing NPC-associated factors, i.e., Slx5/Slx8, which were previously found to contribute to damage relocation to the nuclear periphery[22]. Remarkably, inactivation of either of these two factors caused a decrease in *HSP104* co-fractionation with the nuclear pore complex (Fig. 4g, Supplementary Fig. 4d).

Altogether, this body of evidence demonstrates a requirement for the SUMOylation pathway in mediating R-loop relocation and suggests that the anchoring of R-loop-forming genes at nuclear pore complexes involves interactions between R-loop-bound, SUMOylated RPA complexes and NPC SUMO-interaction motifs.

## Gene repositioning to the nuclear pore complex has a protective effect against R-loop toxicity

We next wondered whether relocation of hybrid-forming genes at NPCs could affect R-loop fate and genetic stability. To this aim, we first assessed the fitness of double mutants combining the hybrid- accumulating *tho* mutation and the inactivation of the different factors uncovered here as mediating R-loop repositioning, i.e., the ssDNA-binding complex RPA, the SUMO-ligase Mms21 and the NPC-associated SUMO-interacting factor Slx8. Growth assays revealed a synergic growth defect of *tho rfa1-D228Y*, *tho mms21-11* and *tho slx8* double mutants as compared to each single inactivation at 25 °C or 30 °C (Fig. 5a). These genetic interactions were further exacerbated at 37 °C for *tho rfa1-D228Y* and *tho mms21-11* mutants (Fig. 5a). In contrast, simultaneous loss-of-function of the two SUMO-ligases Siz1 and Siz2, which detectably impacts cell fitness (Supplementary Fig. 4e) but does not impair R-loop relocation (Supplementary Fig. 4c), did not aggravate the growth defects of the R-loop-forming *tho* mutant (Supplementary Fig. 4e). Since none of these mutations appears to prevent *HSP104* induction (Supplementary Fig. 4f-h), these observations point to a protective effect of the R-loop relocation pathway in conditions of R-loop accumulation.

To further investigate the consequences of NPC association on R-loop metabolism, we set out to monitor R-loop-dependent genetic instability upon persistent peripheral localization of a hybrid-forming locus. To this aim, we took advantage of the presence of LexA-binding sites (LexA-BS) downstream of the *YAT1* reporter system used above (Fig. 1g), and co-expressed a fusion of the bacterial LexA protein to the basket nucleoporin Nup60 to tether the locus to the nuclear pore complex (Fig. 5b), as previously achieved[32]. Microscopy analyses validated high levels of peripheral localization for *YAT1* in LexA-Nup60-expressing cells, independently of its transcriptional status (Supplementary Fig. 4i, red bars), confirming the efficiency of the tethering

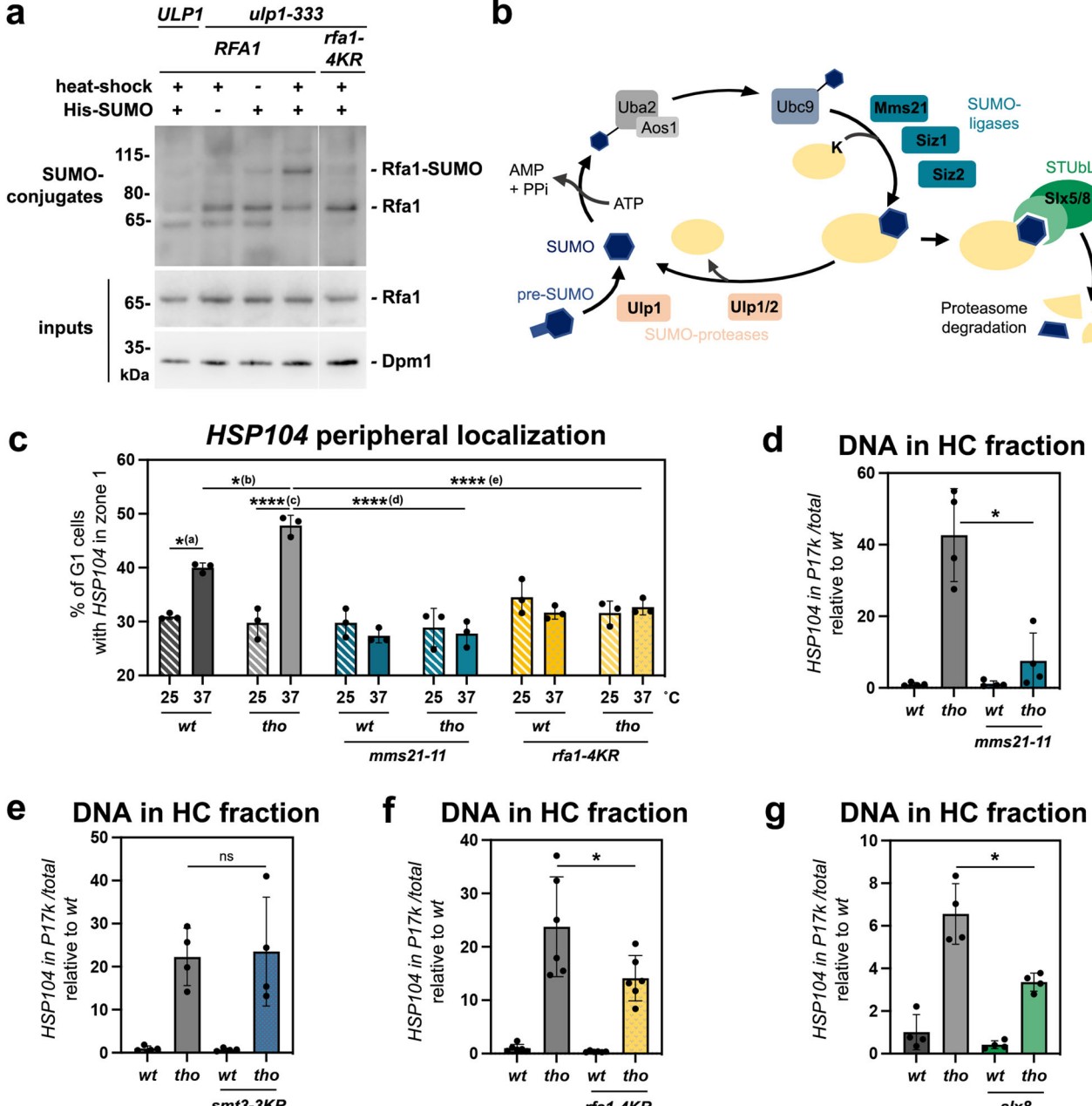

**Fig. 4 | The SUMOylation pathway is involved in R-loop-dependent repositioning to NPCs. a** Western blot detection of Rfa1 in input fractions (bottom panel) or purified SUMO-conjugates (top panel) obtained from the indicated strains. Cells carrying the His-SMT3 (His-SUMO) construct as indicated were grown at 25 °C or heat shocked at 37 °C for 15 min (heat shock). The positions of Rfa1 species are indicated, as well as molecular weights (kDa, kilodaltons). The apparent molecular mass of the Rfa1-SUMO species is ~90 kDa, which is consistent with mono-SUMOylation (Rfa1: 70 kDa; apparent molecular weight of SUMO: 15–20 kDa). One representative experiment (out of three) is displayed; the two other biological replicates are featured in Supplementary Fig. 4a, b. Dpm1 is used as a loading control for input fractions. **b** Overview of the components of the SUMO pathway in *S. cerevisiae*. **c** Fraction of G1 cells (%) showing *HSP104* in zone 1 (mean ± SD, *n* = 3 independent experiments) in the indicated strains grown at 25 °C or heat shocked at 37 °C for 15 min. Statistical test: two-sided Fisher's exact test; *P*-values were calculated on the total number of counted cells (between 287 and 409 cells/condition); (a), $p = 1.59 \times 10^{-2}$; (b), $p = 2.45 \times 10^{-2}$; (c), $p = 4.96 \times 10^{-7}$; (d), $p = 2.84 \times 10^{-8}$; (e), $p = 4.43 \times 10^{-5}$. Values for *wt* and *tho* mutant cells (in gray) are the same as in Fig. 3h. **d**–**g** qPCR-based quantification of the amount of DNA from the *HSP104* locus in heavy chromatin (HC) fractions from the indicated strains heat shocked at 37 °C for 15 min (% of *HSP104* in P17K relative to total [S17K + P17K]; mean ± SD, *n* = 4 independent experiments for panels **d**, **e**, **g**, *n* = 6 independent experiments for panel **f**, relative to *wt*). Statistical test: two-sided Mann-Whitney-Wilcoxon test; **d**, *$p = 2.86 \times 10^{-2}$; **f**, **$p = 4.11 \times 10^{-2}$; **g**, *$p = 2.86 \times 10^{-2}$. Note that experiments involving *mms21-11*, *smt3-3KR* or *rfa1-4KR* mutants are performed with isogenic W303 derivatives, in which the co-fractionation phenotype is reproducibly more pronounced than in other genetic backgrounds. Source data are provided as a Source Data file.

system. As expected from our previous observations (Fig. 1i), transcriptional activation (galactose medium) induced increased localization of the reporter to the nuclear envelope in wild-type cells expressing the LexA protein alone, a phenotype further enhanced in the *tho* mutant (Supplementary Fig. 4i, blue bars). In this context,

introducing the tethering construct did not further increase the peripheral localization of the reporter in *tho* cells. To circumvent this issue, we thereby performed recombination assays upon low levels of transcription of the reporter (glucose-containing medium). In these conditions, cells displayed low but detectable levels of R-loop-

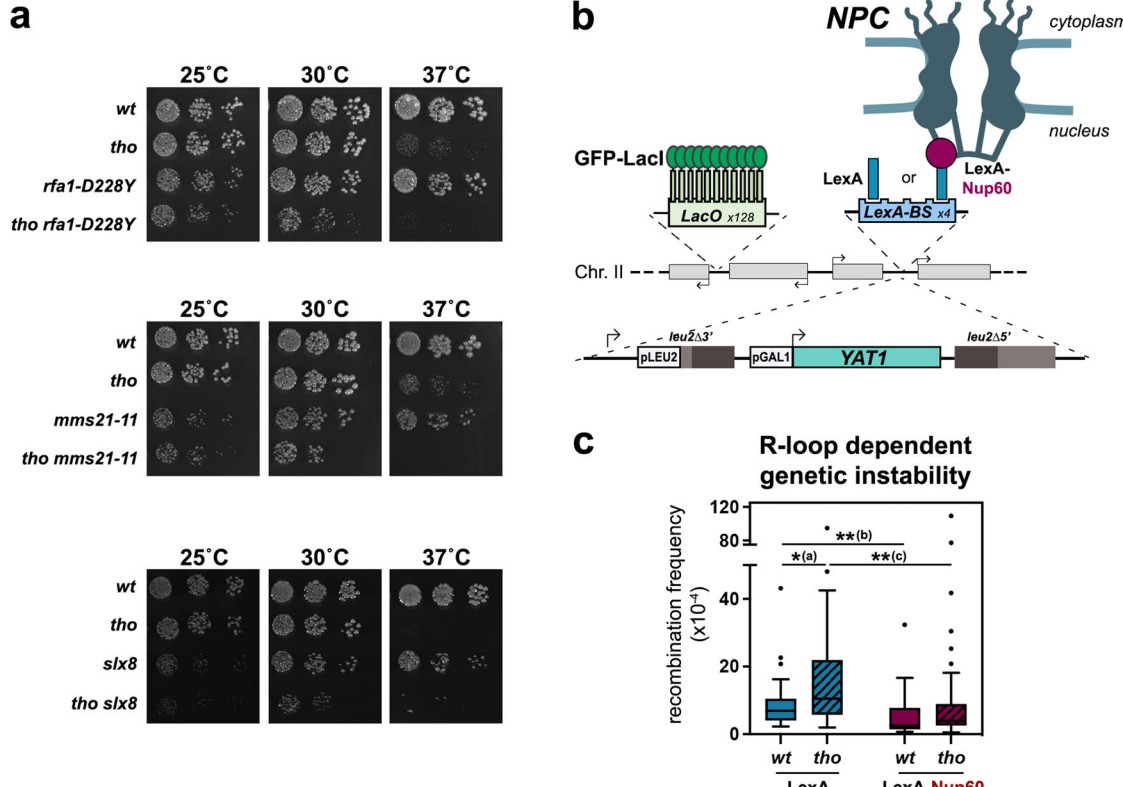

**Fig. 5 | NPC association alleviates R-loop toxicity. a** Serial dilutions of the indicated strains were grown at the indicated temperatures (25 °C, 30 °C, 37 °C) on rich medium (YPD). **b** Principle of the tethering assay. The intronless version of the *YAT1* transgene, under the control of the *GAL1-10* promoter, is flanked by direct *leu2* repeats to allow quantification of R-loop-dependent recombination events. The reporter is integrated at the chromosome II *GAL* locus, which also contains an array of Lac operator (LacO) repeats for microscopy visualization and LexA-binding sites. Expression of the LexA-Nup60 fusion ensures the permanent tethering of the locus to NPCs. **c** Recombination frequencies were calculated for the indicated strains as described in Methods (fraction of Leu+ prototrophs, ×10$^{-4}$; $n$ = 31 for *wt* LexA, $n$ = 32 for *tho* LexA, $n$ = 33 for *wt* LexA-Nup60, $n$ = 35 for *tho* LexA-Nup60; $n$ refers to biologically independent cultures). Glucose treatment was achieved for 1 h following 6 h growth in glycerol-lactate medium. Box-plots are defined as above (Fig. 1d–f). Statistical test: two-sided Mann-Whitney-Wilcoxon test; (a), $p = 2.57 \times 10^{-2}$; (b), $p = 1.2 \times 10^{-3}$; (c), $p = 1.6 \times 10^{-3}$. Source data are provided as a Source Data file.

dependent recombination, as this phenotype could be enhanced in the *tho* mutant (Fig. 5c, in blue). Strikingly, recombination frequencies were significantly reduced when the gene was persistently attached to the nuclear pore complex (Fig. 5c, in red). Altogether, these observations support a protective effect of the NPC against R-loop dependent genetic instability.

## Discussion

In this study, we demonstrate that the co-transcriptional formation of R-loops can trigger the relocation of highly expressed or inducible genes to NPCs (Figs. 1–2). Our data support a model in which the coating of ssDNA by RPA and Mms21-dependent mono-SUMOylation events allow the sensing of R-loops and their association with nuclear pore complexes, where they are bound by SIM-containing NPC partners (Figs. 3–4). Proximity to the NPC would then alleviate R-loop accumulation and/or genotoxicity (Fig. 5), thus allowing to maintain high transcription levels while preserving genetic integrity (Supplementary Fig. 5).

Our genome-wide analyses of NPC-associated genes first unveiled a correlation between nuclear pore complex association and the propensity to form R-loops (Figs. 1b–f, 2d, e). The generation of high-resolution R-loop maps, especially in conditions of stress or metabolic shift in which loci relocation is detected, could provide further insights into the mechanisms of hybrid sensing and NPC targeting. In this respect, recently described RNase H-based R-loop capture methods[39,54] shall provide greater sensitivity in mapping short-lived DNA:RNA hybrids, particularly upon HS (Fig. 2a; Supplementary Fig. 2a) or in *tho*

mutants where transcription is lower at certain loci[41]. Together with the observation that highly expressed, R-loop-deprived intron-containing genes do not associate with NPCs (Fig. 1), our findings support the idea that NPC association is driven by R-loop patterns rather than transcriptional activity. Consistently, interfering with R-loop formation at model loci in *cis*, through the insertion of an intron (Fig. 1i) or in *trans*, via RNAse H over-expression (Fig. 2c; Supplementary Fig. 2b, h, i), similarly abrogated relocation at nuclear pore complexes without down-regulating transcription. Although indirect effects of splicing or RNase H activity cannot be excluded, these observations support that R-loops are the primary causes of NPC repositioning. Whether R-loop formation also partakes in the multiple situations where transcriptional activation drives NPC relocation[74] yet remains to be investigated.

Our genetic dissection of the signals and pathways underlying the relocation of hybrid-forming genes further supports that R-loop gating does not occur through TREX-2 dependent recruitment to NPCs (Fig. 3b), DNA damage formation (Fig. 3c) or replication impairment (Fig. 3d). While we cannot formally exclude that DSBs could arise at R-loop-forming loci independently of the nucleases assessed here (Fig. 3c), it should be noted that the peripheral localization of DNA breaks requires Siz2-dependent poly-SUMOylation[35], in contrast with R-loop repositioning, which specifically involves Mms21-dependent mono-SUMOylation (Fig. 4c–e; Supplementary Fig. 4c). Our data rather suggest that it is the direct sensing of the R-loop structure itself through the ssDNA-binding RPA complex that mediates NPC relocation. Indeed, RPA is detected onto R-loop forming genes with a

dependency for high transcription levels and DNA-binding activity (Fig. 3e), in the absence of replication (Supplementary Fig. 3a, b), and decreased RPA recruitment hinders NPC association (Fig. 3f–h). The labeling of RPA-coated R-loops by Mms21-dependent mono-SUMOylation may further distinguish them from other types of ssDNA-exposing structures, providing competence for binding to the nuclear pore complex via SIM-containing NPC partners (Fig. 4g; Supplementary Fig. 4d). Different mechanisms could restrict the activity of this SUMO-ligase to R-loop-forming regions. Mms21 is part of the cohesin-like Smc5/6 complex, which was shown to be recruited to DNA in a R-loop-dependent manner in the context of Epstein-Barr Virus infection[75], this association being possibly fostered by the increased affinity of this complex for branched DNA structures in vitro[76]. Moreover, Mms21 enzymatic activity has been shown to be enhanced by ssDNA binding in vitro[77]. In line with the *modus operandi* of SUMO-ligases, which typically lack substrate specificity and trigger protein group SUMOylation once recruited[34], Mms21 could thereby target several distinct, yet-to-be-identified R-loop-bound factors at hybrid-forming loci. The fact that RPA SUMOylation increases concomitantly with R-loop gating (Fig. 4a), and that loss of Rfa1 SUMOylation substantially diminishes NPC association (Fig. 4c, f) suggests that RPA is one of the main targets in this process. In the future, assessing the SUMOylation of RPA or other R-loop-associated factors in distinct genetic situations leading to hybrid accumulation shall shed light on the pattern of modifications specifically leading to R-loop repositioning. Remarkably, Mms21-dependent mono-SUMOylation is also required for the relocation of replication forks spanning triplet nucleotide repeat regions (TNRs), where stalled intermediates associate with NPCs prior to damage formation and checkpoint activation, thus alleviating repeat instability[18,20], and which also reportedly form R-loops[59,78]. RPA SUMOylation has also been detected in senescent telomerase-negative cells, concomitantly with the NPC repositioning of eroded telomeres[36]. Although the R-loop gating pathway proceeds independently from replication (Fig. 3d), it thus shares common factors with these alternative relocation processes, suggesting the existence of convergent mechanisms ensuring the detection and the labeling of non-canonical ssDNA-containing structures.

Repositioning to NPCs is generally described as beneficial for gene expression and the maintenance of genome integrity. Tethering experiments indicate that the proximity to nuclear pore complexes indeed alleviates R-loop accumulation[46] and R-loop-dependent recombination (Fig. 5c). In contrast, preventing R-loop repositioning by interfering with the RPA/Mms21 pathway gives rise to synthetic lethality in hybrid-forming *tho* mutants (Fig. 5a). Similarly, inactivation of the Nse1 subunit of the Smc5/6 complex enhances the growth defect of R-loop-accumulating RNase H mutants (*rnh1Δ rnh201Δ*[79]). It remains to be determined whether decreased cell fitness is actually caused by excess R-loop accumulation in these different situations. Remarkably, mutants of the Smc5/6 complex and of the Nup84 complex, which supposedly anchors Slx5/8 to NPCs[16], similarly display increased levels of R-loops in yeast[45,79]. How the NPC environment ultimately influences R-loop fate also requires future investigation. The vicinity to the NPC could allow the mRNA to engage more rapidly in its export path, facilitating its eviction from the transcription site and thus preventing R-loop formation, as previously proposed[46]. Alternatively, the association with NPCs could give access to dedicated R-loop-resolving enzymes, or other factors protecting these structures from breakage. However, none of our previous proteomic analyses of nuclear pore complexes identified interactors related to R-loop metabolism, at least in wild-type cells[70,71]. Finally, recruitment to NPCs could allow the removal of R-loop-bound proteins stabilizing the hybrids or promoting their processing into genotoxic intermediates. In this respect, Ulp1-mediated deSUMOylation, Slx5/8-dependent ubiquitination and degradation by the proteasome, which also resides at the nuclear basket[80], could ensure such clearance events. Whether RPA removal

from R-loops requires its proteolysis and further destabilizes these three-stranded structures at NPCs remains to be explored. In a scenario combining these different models, a "pioneering" R-loop would form during early transcription cycles and rapidly engage the induced gene in NPC association. This event would both allow the local destabilization of the R-loop and prevent the subsequent formation of additional DNA:RNA hybrids at this locus (Supplementary Fig. 5). R-loop-dependent repositioning would thus be particularly critical for inducible genes undergoing several rounds of transcription in a short timeframe, ensuring the high rate of RNA production necessary to sustain viability.

## Methods

### Yeast strains and plasmids

All *S. cerevisiae* yeast strains used in this study (listed in Supplementary Table 2) were obtained by homologous recombination and/or successive crosses. W303 derivatives are *RAD5*+. The construction of the plasmids used in this study (listed in Supplementary Table 3) was performed using standard PCR-based molecular cloning techniques and was checked by sequencing. Yeast strains and plasmids generated in this study are available upon request, without restrictions. Cells were grown at the indicated temperature in standard yeast extract peptone dextrose (YPD) or synthetic complete (SC) medium supplemented with the required nutrients. For heat shock, cells were grown at 25 °C in the appropriate medium to $OD_{600nm} = 0.5$–0.6, quickly shifted at 37 °C by addition of one volume of medium prewarmed at 49 °C or one half-volume of medium pre-warmed at 61 °C and further maintained at 37 °C for 15 min in a water-bath. For experiments involving *GAL* promoter induction, cells were grown at 30 °C in glycerol-lactate (GGL: 0.17% YNB, 0.5% ammonium sulfate, 0.05% glucose, 2% lactate and 2% glycerol) supplemented with the required nutrients prior to induction with glucose or galactose (2%) for 5 h. For experiments involving tet-OFF scRNase H1 induction, cells transformed with the RNH1-overexpressing construct were grown in SC medium supplemented with doxycycline (5 μg/mL, Sigma) and induction was achieved by transferring cells in fresh medium without doxycycline for 16 h. G1 cell cycle arrest was triggered at 25 °C by three sequential additions of alpha-factor (2 μg/ml, Biotem) spaced by 1 h, prior to heat shock; effective synchronization was verified by microscopy observation of cell morphology and flow cytometry. Growth assays were performed by spotting serial dilutions of exponentially growing cells on solid medium and incubating the plates at the indicated temperatures for 2 days.

### Chromatin immunoprecipitation

For Nic96 ChIP-seq, cells were crosslinked for 10 min with 1% formaldehyde at RT under agitation. Excess formaldehyde was quenched with glycine 0.25 M, cells were washed with cold TBS (20 mM Tris, 150 mM NaCl), and pellets were frozen and conserved at −80 °C. Cell pellets were resuspended in lysis buffer (50 mM Hepes pH7.5, 140 mM NaCl, 1 mM EDTA, 1% Triton X-100, 0.1% Na-deoxycholate) supplemented with 1 mM PMSF and protease inhibitors (cOmplete Tablet, Roche) and lysed by beads-beating (Precellys® 24, Bertin). The lysate was sonicated with a Bioruptor (Diagenode) and centrifuged at $2000 \times g$ for 15 min at 4 °C. The supernatant was incubated with anti-Myc (9E10, Santa Cruz Biotechnology, RRID:AB_627268) on a rotating wheel overnight at 4 °C. Dynabeads Protein G (Thermo Fisher Scientific) were equilibrated in lysis buffer and 30 μl were added per sample and incubated on a rotating wheel for 2.5 h at 4 °C. Beads washes were as follows: twice with lysis buffer, twice with lysis buffer supplemented with 360 mM NaCl, twice with wash buffer (10 mM Tris-HCl pH8, 0.25 M LiCl, 0.5% IGEPAL, 1 mM EDTA, 0.1% Na-deoxycholate) and once with TE (10 mM Tris-HCl pH8, 1 mM EDTA). Antibodies were uncoupled from beads with 100 μl of Elution Buffer (50 mM Tris-HCl pH8, 10 mM EDTA, 1% SDS) for 10 min at 65 °C. Decrosslinking was performed at

65 °C overnight. After 30 min of RNase A treatment (20 μg, Roche), proteins were digested by the addition of 100 μg of Proteinase K (Sigma) and incubation for 1.5 h at 37 °C. DNA was purified using the kit InnuPrep PCRpure (Eurobio) and eluted into 35 μl of $H_2O$ prior to library preparation and deep-sequencing.

For Nic96 ChIP-qPCR, cells (20 OD) were cross-linked with formaldehyde 1% for 10 min at the same temperature used for the growth, in the presence of potassium phosphate 100 mM pH 7.5. Excess formaldehyde was quenched with glycine 0,27 M, cells were washed with cold TBS, and pellets were frozen in liquid nitrogen. Frozen cells were lysed by bead beating in 1 mL of lysis buffer (50 mM HEPES pH7.4, 140 mM NaCl, 1 mM EDTA, 1% Triton X-100, 0.1% Na deoxycholate, 4 μg/mL pepstatin A, 180 μg/mL PMSF and protease inhibitor, as above). Chromatin sonication was achieved using a Bioruptor (Diagenode) and the fragmented chromatin was recovered in the supernatant after a 5 min 2500 × g centrifugation at 4 °C. An aliquot was taken as an input fraction (2%) and the remaining sample was mixed overnight at 4 °C with 10 μL of anti-Myc (9E10, Santa Cruz Biotechnology Cat#sc-40, RRID:AB_627268; final concentration 1:100) in the presence of 0.5% (w/v) bovine serum albumin (Sigma) and 47.5 μg/mL salmon testes DNA (Sigma). Dynabeads Protein G (10 μL, Thermo Fisher Scientific) were then pre-coated for 1 h in blocking buffer (lysis buffer as above containing 0.5% (w/v) bovine serum albumin [Sigma] and 47.5 μg/mL salmon testes DNA [Sigma]) and mixed with the immunoprecipitation mixtures for 1 h. Beads washes were as follows: twice with lysis buffer; twice with lysis buffer supplemented with 360 mM NaCl; twice with 10 mM Tris pH 8, 250 mM LiCl, 0.5% Nonidet-P40, 0.5% deoxycholate, 1 mM EDTA, and once with 10 mM Tris–HCl pH 8, 1 mM EDTA. Immunoprecipitated complexes were eluted for 10 min at 65 °C in 100 μL 50 mM Tris pH 8, 10 mM EDTA, 1% SDS, and deproteinized with 16 μg proteinase K in the presence of 250 mM NaCl for 1 h at 42 °C and for 30 min at 65 °C. Input and immunoprecipitated DNAs were purified with the QIAquick DNA purification kit (Qiagen) and further quantified by real-time PCR.

RNase H ChIP was performed with the same protocol as the Nic96 ChIP with the exception that the fragmented chromatin was mixed with 25 μL of Anti-FLAG® M2-Agarose beads (Sigma) overnight at 4 °C, prior to the washes.

RPA ChIP was achieved as above, with the following modifications. The fragmented chromatin (obtained from 25 OD of cells) was mixed overnight at 4 °C with either anti-RPA (Agrisera Cat# AS07 214; RRID:AB_1031803) or a pre-immune control serum (5 μL each; final concentration 1:200), in the presence of 0.5% (w/v) bovine serum albumin (Sigma) and 47.5 μg/mL salmon testes DNA (Sigma). Input and immunoprecipitated DNAs were purified with the Nucleospin Gel and PCR Clean-up kit (Macherey Nagel) prior to real-time PCR.

### Genome-wide sequencing
DNA libraries were prepared using NEBNext Ultra DNA Library Prep Kit for Illumina (New England Biolabs) according to the manufacturer's specifications. Each library was quantified on Qubit with Qubit dsDNA HS Assay Kit (Thermo Fisher Scientific) and size distribution was examined on the Bioanalyzer 2100 with High Sensitivity DNA chip (Agilent), to ensure that the samples have the proper size, no adaptor contamination and to estimate sample molarity. Each library was diluted to 1 nM and then pulled together at equimolar ratio. Libraries were denatured according to the manufacturer's instruction and sequenced on a mid-output flow cell (130 M clusters) using the Next-Seq 500/550 Mid Output kit v2.5 150 cycles kit (Illumina), in paired-end 75/75 nt mode, according to the manufacturer's instructions.

### Bioinformatic analyses
Highly expressed intronless and intron-containing gene groups were defined as before[42], with the exception that genes encompassing repeated sequences leading to ambiguous mapping were excluded from the analysis (see Supplementary Table 1 for the list of the 71 intronless genes and 80 intron-containing genes considered here).

Nic96 ChIP-Seq data quality was assessed using FastQC[81]. Paired-end reads were mapped to *S. cerevisiae* genome (2011, SacCer3) with Bowtie2[82], allowing only perfect matches. Duplicated reads were removed using SAMTools rmdup[83] to obtain Binary Alignment Mapped (BAM) file. Normalized fragments per kilobase per million mapped fragments (FPKM) were subjected to peaks calling using MACS2[84] with a q-value < 0.05. Peak annotation was done with the BEDTools ClosestBed[85] by determining the closest genomic feature to the summit position of the MACS2 peak. Normalized bigwig files (subtracting the no tag ChIP from the Nic96 ChIP) and heatmaps were obtained using deepTools2[86]. Read profiles were visualized with the Integrative Genomics Viewer (IGV)[87]. Nup170 or Nup157 enrichments were represented as the average log2 (IP/input) for all the probes covering a given genomic feature. For RPA ChIP-seq, normalized bigwig files (subtracting the input from the RPA ChIP) and heatmaps were obtained using deepTools2[86].

The following calculations were used to evaluate biases in base content: GC skew = (G-C)/(G + C); AT skew = (A-T)/(A + T). Nic96 (this study), RPA[62] and R-loop[38] occupancies were determined by integrating ChIP- or DRIP-seq counts over transcription units (Fig. 1d; Supplementary Figs. 1d, 3b).

### Chromatin fractionation
Differential chromatin fractionation was performed as previously described[52]. Cells (20 OD) were cross-linked with formaldehyde 1% for 10 min at 37 °C in the presence of potassium phosphate 100 mM pH 7.5. Excess formaldehyde was quenched with glycine 0.27 M, cells were washed with cold TBS, and pellets were frozen in liquid nitrogen. Cell pellets were resuspended in Lysis buffer (50 mM Hepes pH 7.5; 150 mM NaCl; 1 mM EDTA pH8; 1% Triton X-100; 0.1% Na deoxycholate; 0.1% NP40; 0.1% SDS; 4 μg/mL pepstatin A; 180 μg/mL PMSF) and lysed by bead beating using a Fastprep (QBIOGENE). Following a centrifugation for 10 min at 12,000 × g in a bench centrifuge at 4 °C, the chromatin pellet was resuspended in lysis buffer and sonicated with a Bioruptor (Diagenode). The lysate was then centrifuged 10 min at 300 × g to remove cellular debris and 1 mL of the supernatant, containing the chromatin, was further centrifuged for 10 min at 17,000 × g to isolate the heavy chromatin fraction. The pellet was washed in lysis buffer and resuspended in 100 μL elution buffer (50 mM Tris pH8; 10 mM EDTA; 1% SDS). To decrosslink, 50 μg of proteinase K were added to 100 μL of the supernatant (S17K) and the resuspended pellet (P17K), and the samples were incubated 30 min at 37 °C and 1 h at 65 °C. DNA was purified with the QIAquick DNA purification kit (Qiagen) according to the manufacturer's instructions and quantified by real-time PCR.

### DNA:RNA hybrid detection
DNA:RNA hybrid immunoprecipitation (DRIP) was performed using the S9.6 DNA:RNA hybrid-specific monoclonal antibody according to a published procedure[38,88], with the following modifications. Briefly, genomic DNA was phenol-extracted from 25 OD of exponentially growing yeast cells and isolated by ethanol precipitation. 40 μg of purified DNA were digested by a cocktail of restriction enzymes (EcoRI, HindIII, XbaI, SspI, BsrGI; FastDigest enzymes; Thermo Fisher Scientific) for 30 min at 37 °C in a total volume of 100 μL. Specificity of the DRIP signal was determined by including 20 units of RNase H (New England Biolabs) in the digestion reaction. An aliquot of the digested DNA was taken as an input fraction (5%); the remaining sample was diluted fourfold with TBS 0,1% Tween 20 and mixed overnight at 4 °C in the presence of 0,3 μg of S9.6 purified antibody[42]. Immunoprecipitated DNA fragments were further captured on protein G Sepharose beads (GE Healthcare) mixed with the immunoprecipitation mixtures for 1 h at 4 °C. Beads were then washed 5 times with TBS Tween buffer and the immunoprecipitated hybrids were eluted for 20 min at 65 °C in

100 µL 50 mM Tris pH 8, 10 mM EDTA, 1% SDS and then deproteinized with 16 µg proteinase K in the presence of 250 mM NaCl for 1 h at 42 °C and for 30 min at 65 °C. Input and immunoprecipitated DNAs were purified with the QIAquick DNA purification kit (Qiagen) according to the manufacturer's instructions and further quantified by real-time PCR.

## RNA extraction
Total RNAs were purified from 10 OD of cells using the Nucleospin RNAII kit (Macherey Nagel) according to the manufacturer's instructions. For quantitative PCR (RT-qPCR), RNAs were reverse-transcribed using random hexamers (P(dN)6, Roche) and Superscript II reverse transcriptase (Thermo Fisher Scientific).

## Nucleic acid analyses
DNA amounts in ChIP, DRIP, chromatin fractionation, and cDNAs samples were quantified by real-time PCR with a LightCycler 480 system (Roche) using SYBR Green incorporation according to the manufacturer's instructions. For IP experiments, the amount of DNA in the immunoprecipitated fraction was divided by the amount detected in the input to evaluate the percentage of immunoprecipitation (% of IP). For RPA ChIP, the % of IP obtained using a control pre-immune serum was subtracted to the one from anti-RPA IPs. For differential chromatin fractionation, the amounts of DNA in the P17K and S17K fractions were quantified; the % of DNA in the heavy chromatin fraction was calculated as P17K/(P17K + S17K). Measurements of cDNA levels following RT-qPCR were normalised to *ACT1* mRNAs. The sequences of the primers used in this study are listed in Supplementary Table 4.

## Protein analyses
SUMO-conjugates were isolated from yeast cells expressing a polyhistidine-tagged version of SUMO using nickel agarose denaturing chromatography as previously described[70], starting from 50 mL of exponentially growing cells (OD$_{600}$ = 0.5−1). Protein samples were separated on 4−12% precast SDS-PAGE gels (Thermo Fisher Scientific). Proteins were further detected by western-blot following transfer to PVDF membranes. The following validated antibodies were used: anti-RPA polyclonal antibody (same as for ChIP), 1:1000; anti-Dpm1 monoclonal antibody (ThermoFisher Scientific Cat#A6429; RRID: AB_2536204), 1:1000; anti-rabbit HRP secondary antibody (Jackson Immunoresearch Cat#711-035-152; RRID:AB_10015282), 1:5000. Images were acquired using chemiluminescent reagents (Supersignal, Thermo Fisher Scientific) with a ChemiDoc MP Imaging System (Bio-Rad).

## Live cell imaging
Exponentially growing cells were harvested by centrifugation and mounted on slides for imaging. Live cell images were acquired using a wide-field inverted microscope (Leica DMI-6000B) equipped with Adaptive Focus Control to eliminate Z drift, a 100×/1.4 NA immersion objective with a Prior NanoScanZ Nanopositioning Piezo Z Stage System, a CMOS camera (ORCA-Flash4.0; Hamamatsu) and a solid-state light source (SpectraX, Lumencore), piloted by the MetaMorph software (Molecular Device). For GFP-mCherry two-color images, 19 focal steps of 0.20 µm were acquired sequentially for GFP and mCherry with an exposure time of 100 ms using solid-state 475- and 575-nm diodes and appropriate filters (GFP-mCherry filter; excitation: double BP, 450−490/550−590 nm and dichroic double BP 500−550/600−665 nm; Chroma Technology Corp.). Processing was achieved using the ImageJ software (NIH). Images shown are z projections of z-stack images. Image analysis was realized with the FIJI software[89]. Distances between loci and nuclear envelope were measured using either the PointPicker plugin[90] or a home-made macro. The G1 stage was determined on the basis of cellular morphology (unbudded cells).

## Hyper-recombination assay
Independent clones were individually resuspended in 1 mL glycerol-lactate medium, grown for at least 2 h at 30 °C and then induced with glucose or galactose (2%) for the indicated period of time. Cells were resuspended in water, appropriate dilutions were plated on SC medium lacking leucine to estimate the number of Leu+ recombinants, or complete medium to estimate cell survival, and plates were incubated for 2 days at 30 °C. Hyper-recombination rates were defined as the proportion of Leu+ prototrophs.

## Quantification and statistical analysis
The following statistical tests were used to evaluate the statistical differences between strains/conditions: two-sided Mann-Whitney-Wilcoxon rank sum test, to assess (i) Nic96, Nup170, Nup157, Rnh1 or RPA occupancies in ChIP-seq, ChIP-chip or ChIP-qPCR analyses, (ii) gene abundances in chromatin fractionation assays, and (iii) recombination levels; two-sided Fisher's exact test, to compare the fraction of cells showing localization of the locus of interest in zone 1 or at NPC clusters in microscopy experiments. Box-plots were represented according to Tukey's definition using Prism v8.0.2. Standard conventions for symbols indicating statistical significance are used ($*p < 0.05$; $**p < 0.01$; $***p < 0.001$; $****p < 0.0001$; ns, $p > 0.05$, not significant) and exact $p$-values are provided whenever possible.

## Data availability
The complete sequencing data generated during this study (Nic96 ChIP-seq) have been deposited in NCBI's Gene Expression Omnibus (GEO) and are accessible through GEO Series accession number GSE225324. Sequencing data were mapped to the SacCer3 version of the budding yeast genome. RNase H CRAC datasets[39] are available through the GEO Series accession number GSE195936. DRIP-seq data[38] were retrieved from the Sequence Read Archive with the accession number SRP071346. Nup170/Nup157 Chip-on-chip datasets[47] and RPA ChIP-seq data from control, alpha-factor-arrested cells[62] were obtained from GEO through accession numbers GSE36795 and GSE182203. Source data are provided with this paper and include the numeric data supporting all featured graphs, as well as uncropped scans from blot images and drop assays. All the other data generated in this study are provided in this article and its Supplementary Information files. Source data are provided with this paper.

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

## Acknowledgements

We are very grateful to Andrès Aguilera, Catherine Dargemont, Catherine Freudenreich, Doug Koshland, Stéphane Marcand, Françoise Stutz and Xiaolan Zhao for sharing yeast strains and plasmids; to Florent Dobé, Léa Doré and Simon Morin for technical help; to members of the ImagoSeine@IJM facility (Université Paris Cité, CNRS, Institut Jacques Monod) for flow cytometry analyses; to Christelle Cayrou for help with the NGS experiments; to Nadine Carbuccia and Emilie Mamessier for NGS facility access; and to Umberto Aiello, Anna Babour, Axel Cournac, Gilles Fischer, Manuel Mendoza, Rodney Rothstein, Vincent Vanoosthuyse, Michel Werner and members of the Palancade and Libri lab for fruitful discussions. This work was supported by Agence Nationale pour la Recherche (ANR-18-CE12-0003 to B.P., K.D. and V.G.; ANR-21-CE12-0029 to M.R. and B.P.; ANR-21-CE12-0040 to D.L. and B.P.), Fondation

ARC pour la recherche contre le Cancer (projet ARC, to B.P.), Ligue Nationale contre le Cancer (comité de Paris, to B.P.), the Université Paris Cité IdEx (ANR-18-IDEX-0001, to B.P.), the BioSPC PhD program (to A.P., R.M.M., and M.Z.), the Fondation pour la Recherche Médicale (grant number FDT202106013057, to A.P.) and the EUR G.E.N.E. (ANR-17-EURE-0013, to A.P.).

## Author contributions

Conceptualization: A.P., V.G., K.D., and B.P.; Methodology: A.P., M.D., M.Z., R.M.M., M.N.S., V.G., K.D., and B.P.; Investigation: A.P., M.D., C.B., M.Z., R.M.M., M.R., C.G., O.L., D.L., M.N.S., V.G., K.D., and B.P.; Formal analysis: M.D. and B.P.; Visualization: A.P. and B.P.; Writing (first draft): A.P. and B.P.; Writing (editing): all authors; Funding acquisition: A.P., M.R., D.L., V.G., K.D., and B.P.; Supervision: M.N.S, V.G., K.D., and B.P.

## Competing interests

The authors declare no competing interests.
