## [Peer Review File · Nature Communications]

A R-loop sensing pathway mediates the relocation of transcribed genes to nuclear pore complexesREVIEWER COMMENTS

Reviewer #1 (Remarks to the Author):

In this manuscript by Penzo et al., the authors have examined the potential role of R-loops in mediating the association of highly transcribed genes with the nuclear pore complex (NPC) and the potential role of this association in genome stability. Using genome-wide analyses of Nic96 CHIP data, the authors present data that suggest intronless genes that form R-loop structures displayed a higher propensity to bind NPCs than similarly expressed genes containing introns and depleted of R-loops. To further evaluate that the mechanism by which R-loops direct genes to the NPC, the authors developed an *in vivo* assay to assess R-loop-mediated gene localization. They show that the inducible and highly expressed heat shock (HS) gene, HSP104, forms an R-loop upon induction and that it co-fractionates with NPCs in heavy chromatin isolates. Importantly, the increase in HSP104 DNA in heavy chromatin fractions after heat shock was suppressed by HsRNH1 overexpression in the mutant cells. Other HS genes are also proposed to show R-loop dependent interactions with NPCs. Consistent with these conclusions the authors show that RPA is recruited to HSP104 locus during HS and mutations in RPA that reduce its association with ssDNA also reduce HSP104 locus association with the NE. Furthermore, they show the HS-dependent SUMOylation of the RPA subunit Rfa1 is required for the recruitment of HSP104 locus to the NE. Finally, they present data that suggest NPC recruitment has a protective effect against R-loop toxicity.

Overall, the data presented are of high quality, and the experiments are largely well controlled. This novel work describes a previously unappreciated mechanism for tethering active genes to the nuclear periphery through R-loops that are proposed to mediate the interaction of the cognate gene with the NPC and promote genome stability. These results are an important advance in our knowledge of NPC function, and they will be of general interest to those studying the function of NPCs and, more generally, chromatin structure and stability. There are, however, a few additional experiments that could be performed to strengthen the authors' conclusions.

Points to address:

- 1) The evidence that R-loops represent a feature within specific genes that is necessary for, and directly mediates, their interaction with NPCs is lacking. Nic96 CHIP analysis offers a reasonable measure of direct gene association with the NPCs, and the authors use this assay in the early sections of the Results to show that R-loop containing genes appear to enrich at NPCs. However, this same assay is not used to assess whether the R-loops, formed within the intronless genes targeted (see Fig.1) or specifically within HSP104 and other HS genes, are necessary for the ChIP detected interaction with Nic96 (i.e. NPCs). Instead, more indirect assays, (such as microscopy localization of tagged gene loci to the NE and cofractionation of R-loop containing HS genes with NPC-containing heavy chromatin) are used to assess R-loop function in tethering genes to the NPC. In the early stages of their analysis of R-loop association with the NPC, the authors should also perform Nic96 CHIP analysis on cells where the R-loop is disrupted to determine whether the loss, or reduction, of these structures also disrupts the binding of R-loop containing genes to Nic96. A prediction would be that overproducing HsRNH1 in cells would reduce intronless R-loop gene binding to Nic96/NPCs. The author could focus on their top ranked intronless R-loop genes (Fig.1), or at least examine the HS genes as they are the focus of most analyses in the rest of the paper. Importantly, these data would further support the authors conclusions that their microscopy and cofractionation analysis also reflects changes in the association of R-loop containing HS genes with NPCs, not other components of the NE.
- 2) The authors could also consider an alternative approach of tethering HsRNH1 to the NPC (perhaps by fusing to Nup60) as a method to test whether such a construct would disrupt R-loops at NPCs and reduce gene association. This system could be tested by examining association of the HSP104 locus with the NPC. The ability of an HsRNH1-Nup60 (or other Nup) construct to disrupt the association the HSP104 locus with the NE in microscopy localization assays or heavy chromatin fractions would provide further support for the authors conclusion that these assays are evaluating NPC bound R-

loops.

3) The authors should, in general, provide a clearer description on the data presented in the figures, both through the addition of detail in the figures and the figure legends. For example, in Fig. 2D there is no description of the data shown (e.g. what is shown on the y-axis?). In Fig 4A show mass markers. What is the molecular mass of the Rfa1-SUMO? Please generally reexamine the figures and legends.

4) The conclusion that SUMOylated Rfa1 cannot be detected in WT cells is confounded by reduced levels of Rfa1 observed in WT cells when compared to that in *ulp1-333* cells (see inputs of Fig.4A). This raises the question as to whether Rfa1 is stabilized in *ulp1-333* cells. It would be useful to include loading controls for these input fractions. Moreover, other groups have detected Rfa1 SUMOylation in WT cells using SUMO-His pulldowns (see Fig.2F and S1 of Psakhye and Jencksch. 2012. Cell. 151, 807–820). Establishing the assay for detecting Rfa1-SUMO in WT cells before and after HS would allow the authors to examine the requirement for Mms21 in this event. I would ask the authors to address why they have not pursued this further.

5) There appears to be significantly different percent levels of the HSP104 locus recovered in the mutants between experiments. See Fig.4 panels D-G. Why is this the case? These experiments do not appear to be very reproducible.

6) In Fig.5A, the authors' conclusion that the double mutants *tho rfa1-D228Y* and *tho mms21-11* show growth defects at 25C relative to their single mutant counterparts is not supported by the data. The data at 37C is consistent with their conclusion, with the exception of the *tho slx8* mutant, which shows no difference with the *tho* mutant.

7) In Fig. S1F, please shown significance of differences.

8) I would suggest that the authors use a single term, i.e. 'nuclear pore complex or NPC', when referring to nuclear pore complexes and avoid using terms like 'nuclear pore' or 'pore'.

Reviewer #2 (Remarks to the Author):

In their manuscript "A R-loop sensing pathway mediates the relocation of transcribed genes to nuclear pore complexes", Arianna Penzo and co-authors correlate R-loop accumulation on chromatin with "gene gating" of the respective loci to nuclear pores. They further demonstrate a dependence on the R-loop-sensing protein RPA, which requires its mono-SUMOylation by Mms21 and interaction with SUMO-interacting domain-containing NPC components. They go on to show that localization to the NPC has a favourable effect at R-loop containing genes, reducing R-loop induced recombination and genotoxicity.

I really like this study – quite a bit of the data is correlative, but it is thoroughly done and nicely controlled; the proposed mechanism explains the observations start to finish and is well supported by the data. The authors do not go so far as to claim it, but I suspect that their proposed mechanism – or aspects of it – could well turn out to be common to other gene gating pathways. It also raises interesting questions about how different "types" of R-loops are differentiated (rDNA is very R-loop prone, but apparently safely tucked away in the nucleolus) and how gene localization impacts genotoxicity, which will certainly be addressed in future studies. The manuscript is well written, experimental conditions are explained with sufficient detail in the Figure legends / M&M section, statistical analyses have been performed throughout, and the figures look polished.

The following comments arise from curiosity rather than being meant as a criticism of the work:

1) If I understand it correctly, presence in the HC fraction is specific to heat-shock genes in the mutants under heat shock, which have exceptionally high levels of NPC association. The negative controls (Suppl Fig 2c) include some housekeeping genes and intergenic genes, but have the authors tested any of their model R-loop rich genes, like the intronless YAT1 construct, or the highest-ranking genes based on Nic96-myc occupancy? Would these also be found in a HC fraction (% at nuclear periphery appears to be in a similar range)? Or are heat-shock genes in the mutants under heat shock a special / extreme case?

2) I noticed that the LacO/GFP-LacI signal for the YAT1 construct at the nuclear periphery does not markedly co-localize with the Nup signal (Fig 1g), at least for the two cells shown. This would suggest that the interaction with the NPCs is rather labile while still serving to enrich them in the peripheral zone. There are no example pictures included for the HSP104 locus, but have the authors ever scored the degree of colocalization of GFP-LacI spots with NPCs within the peripheral zone and would this vary with conditions? Just wondering whether this would show a (even) stronger correlation with % in HC, assuming that % in HC reflects genes that are actually bound to an NPC.

3) All the experiments supporting the RPA/SUMO-dependent mechanism were carried out using the HSP104+tho-mutant+heat shock system, which may show the strongest phenotype, but is also slightly artificial. It would seem obvious to also test the effect of at least some mutants along the rfa1/mms21/slx axis on a model R-loop rich gene (e.g. the highest-ranking genes based on Nic96-myc occupancy, or the intronless YAT1 construct). Has this been done?

4) Figure 1b visualizes a rough correlation of Nic96-myc occupancy with levels of R-loops at the locus. Because there appears to be an obvious difference in the pattern of Nic96 across intronless and intron-containing genes – covering the entire transcription unit in the former vs. only the 3' half in the latter (suggesting exclusion from the area that contains the intron) – do any of the R-loop data used for the ranking show a similar pattern (for example, in a metagene plot)? To note, the sites of highest Nic96 signal do not markedly colocalize with the sites of highest RPA signal (compare Figure 1b and Suppl Figure 3b) – for the highest-ranking genes, the pattern even appears to alternate (may be wrong, limited by resolution of Suppl Figure 3b).

Minor points:

Abstract: “preventing R-loop-dependent relocation leads to lethality” – This only applies to tho mutants, which have an exceptionally high R-loop burden. Please clarify.

Results, line 212: I believe reference 62 (Reusswig et al. 2020) was inadvertently dropped at the end of the sentence. It clearly belongs there, and is included in this order, but does not appear in the manuscript before the M&M section.

Discussion, line 376: Areas of triplet repeat expansion have been shown to be associated with R-loops, at least in some cases (PMID: 24787137). The authors may want to include the reference.

Figure 1d and Suppl. 3b: For the “ranked by R-loop levels” score used to sort the genes, the Figure legends simply point to reference 38 (suppl ref 2), Wahba et al. 2016, stating “as previously measured”. While the reference contains the corresponding S1-DRIP-seq data, it does not directly provide a score for R-loop levels for each gene nor any details how such a score would be calculated. I assume this would be the integrated DRIP-seq counts over transcription unit +/- 1000 bp, which are the basis for the heat map sorting in Figure 4A of that reference? Please clarify.

Figure 2d: Since all the other panels are nicely labelled with the experiment type, please also add “Nic96 occupancy” to panel d.

All Figures labelled “DNA in HC fraction” (Figures 2c, 3b,c,d,f, 4d,e,f,g, ... & Supplements): “% relative to wt” does not make any sense – after normalization, it’s just a fold-change in the relative HC fraction association. Please relabel. It’s not quite clear to me why one would want to normalize this to wt in the first place? Just plotting the % HC / total would facilitate comparison to the peripheral localization data.

Figure 5c and Suppl Fig 1g: “Recombination frequency” is missing a “b”.

Reviewer #3 (Remarks to the Author):

In the manuscript by Penzo et al, there is exciting data showing that the formation of R-loops results in the repositioning of the R-loop harboring locus to the nuclear pore. This is elegantly demonstrated using a host of techniques ranging from ChIP-seq to microscopy to genetic manipulation. Using ChiP with the Nic96 protein it is shown that R-loop harboring (largely intronless) genes are found associated with the pore, this is also consistent with other nuclear pore complex members. The authors then show that the Yat1 locus localizes to the nuclear periphery upon transcription induction however it fails to do so when an intron is inserted into the locus, which prevents R-loop accumulation. R-ChIP is employed to demonstrate the R-loops accumulate at heat shock loci and that these loci accumulate in the heavy chromatin fraction of cell extracts (consistent with pore association). Importantly, this effect is enhanced when the THO complex is mutated and R-loops accumulate to even higher levels. The relocalization to the pore is likely independent of DNA damage as it is also seen in G1 arrested cells, as well as in rad2, mlh2 and fcy1 mutants. Interestingly, the presence of RPA is also required for the nuclear repositioning of R-loops. Indeed, in rfa1-D228Y mutants the repositioning of the HSP104 locus is faulty following heat shock in a tho mutant background. The SUMOylation of Rfa1 seems to be critical for the nuclear pore localization of R-loops as an Rfa1-4K mutant as well as mms21 mutations prevent it. Finally it is suggested in the final figure that re-localization of R-loops to nuclear pores aids in promoting genome stability.

Altogether this is a manuscript of very high quality and it is very well written. All experiments are well controlled and the message is clear. I feel that the functional implications of R-loop localization are still poorly understood, although Figure 5 attempts to address this, it is not completely convincing. Nonetheless I feel that this manuscript should be directly accepted for publication and will be interesting for a very general audience. I would have one minor suggestion that could be attempted, however it is not essential.

In Figure 1, it would have been nice to show that RNase H1 overexpression reduces the pore association of certain R-loop containing genes. Again, I understand the caveats associated with such an experiment.

Congratulations on a beautiful manuscript.

Reviewer #1

In this manuscript by Penzo et al., the authors have examined the potential role of R-loops in mediating the association of highly transcribed genes with the nuclear pore complex (NPC) and the potential role of this association in genome stability. Using genome-wide analyses of Nic96 CHIP data, the authors present data that suggest intronless genes that form R-loop structures displayed a higher propensity to bind NPCs than similarly expressed genes containing introns and depleted of R-loops. To further evaluate that the mechanism by which R-loops direct genes to the NPC, the authors developed an in vivo assay to assess R-loop-mediated gene localization. They show that the inducible and highly expressed heat shock (HS) gene, HSP104, forms an R-loop upon induction and that it co-fractionates with NPCs in heavy chromatin isolates. Importantly, the increase in HSP104 DNA in heavy chromatin fractions after heat shock was suppressed by *hsRNH1* overexpression in the mutant cells. Other HS genes are also proposed to show R-loop dependent interactions with NPCs. Consistent with these conclusions the authors show that RPA is recruited to HSP104 locus during HS and mutations in RPA that reduce its association with ssDNA also reduce HSP104 locus association with the NE. Furthermore, they show the HS-dependent SUMOylation of the RPA subunit Rfa1 is required for the recruitment of HSP104 locus to the NE. Finally, they present data that suggest NPC recruitment has a protective effect against R-loop toxicity.

Overall, the data presented are of high quality, and the experiments are largely well controlled. This novel work describes a previously unappreciated mechanism for tethering active genes to the nuclear periphery through R-loops that are proposed to mediate the interaction of the cognate gene with the NPC and promote genome stability. These results are an important advance in our knowledge of NPC function, and they will be of general interest to those studying the function of NPCs and, more generally, chromatin structure and stability. There are, however, a few additional experiments that could be performed to strengthen the authors' conclusions.

We acknowledge Reviewer #1 for this very positive assessment of our work and for their suggestions for improvement. We have now included additional experiments and analyses to address the points they raised, as detailed below.

Points to address:

1) The evidence that R-loops represent a feature within specific genes that is necessary for, and directly mediates, their interaction with NPCs is lacking. Nic96 CHIP analysis offers a reasonable measure of direct gene association with the NPCs, and the authors use this assay in the early sections of the Results to show that R-loop containing genes appear to enrich at NPCs. However, this same assay is not used to assess whether the R-loops, formed within the intronless genes targeted (see Fig.1) or specifically within HSP104 and other HS genes, are necessary for the ChIP detected interaction with Nic96 (i.e. NPCs). Instead, more indirect assays, (such as microscopy localization of tagged gene loci to the NE and cofractionation of R-loop containing HS genes with NPC-containing heavy chromatin) are used to assess R-loop function in tethering genes to the NPC. In the early stages of their analysis of R-loop association with the NPC, the authors should also perform Nic96 CHIP analysis on cells where the R-loop is disrupted to determine whether the loss, or reduction, of these structures also disrupts the binding of R-loop containing genes to Nic96. A prediction would be that overproducing *hsRNH1* in cells would reduce intronless R-loop gene binding to Nic96/NPCs. The author could focus on their top ranked intronless R-loop genes (Fig.1), or at least examine the HS genes as they are the focus of most analyses in the rest of the paper. Importantly, these data would further support the authors conclusions that their microscopy and cofractionation analysis also reflects changes in the association of R-loop containing HS genes with NPCs, not other components of the NE.

As suggested by the Reviewer, we have now interfered with R-loop formation by overexpressing *hsRNH1* in *wt* or R-loop accumulating *tho* cells, and further assessed direct gene association with NPCs through Nic96 ChIP (**Novel Suppl. Fig. 2i**). In line with our model and the Reviewer's prediction, these novel experiments show that R-loop reduction leads to a decreased association of the NPC with several HS loci (*HSP104*, *GSY2*, *PAU17*, *SSA4*), both in *wt* and *tho* cells. Conversely, R-loop induction in *tho* cells leads to increased association of these HS genes with the NPC as compared to *wt*. Altogether, these data thus strengthen our conclusion that R-loop formation is a trigger for repositioning to NPCs. Of note, the Nic96 ChIP signals observed over control or intergenic loci also vary slightly in conditions of altered R-loop levels, albeit in a non-significant manner. This observation likely reflects the proximity of control loci with one of the multiple NPC contact sites scored in our ChIP seq analysis (more than 1000 peaks over the genome, see Fig. 2e), as indicated in the corresponding figure legend.

To provide an additional, independent evidence for the interaction of R-loop-forming genes with the NPC, we also performed microscopy analyses in the *nup133Δ* nucleoporin mutant, which leads to the formation of discrete NPC clusters at the NE, a classical strategy to distinguish NPCs from other components of the NE (**Novel Suppl. Fig. 2e-f**). In this background, relocation of the tagged *HSP104* locus to the NE coincides with its recruitment within NPC clusters (see representative images in **Suppl. Fig. 2e** and quantifications in **Suppl. Fig. 2f**). These data thus confirm that the repositioning

at the nuclear periphery probed in our imaging assays indeed reflects gene association with NPCs, rather than other NE domains.

2) The authors could also consider an alternative approach of tethering HsRNH1 to the NPC (perhaps by fusing to Nup60) as a method to test whether such a construct would disrupt R-loops at NPCs and reduce gene association. This system could be tested by examining association of the HSP104 locus with the NPC. The ability of an HsRNH1-Nup60 (or other Nup) construct to disrupt the association the HSP104 locus with the NE in microscopy localization assays or heavy chromatin fractions would provide further support for the authors conclusion that these assays are evaluating NPC bound R-loops.

Our data support a model in which R-loops act as a trigger for gene repositioning, rather than directly mediating the interaction between transcribed DNA and NPCs. However, to evaluate whether localized R-loop clearance would disrupt gene association, we have followed the Reviewer's suggestion and tethered hsRNH1 to the NPC by fusion to Nup60, as previously described for other proteins (e.g. Ulp1, *Panse et al, 2003* - PMID: 12471376; Sac3, *Fischer et al, 2002* - PMID: 12411502). However, the NPC tethering of the hsRnh1-GFP-Nup60 fusion protein was not homogenous within *tho* mutant cells (see below, **Figure R1, left panel**). Moreover, the NPC-associated levels of this fusion protein were strikingly lower than those of a GFP-tagged version of full-length Nup60 (*right panel*), although both proteins were expressed from *NUP60* natural promoter. It is likely that low expression levels of the fusion protein, together with the presence of endogenous Nup60, precludes its efficient NPC tethering. Since this experiment cannot be performed in the absence of endogenous Nup60, in view of the cohesivity between *nup60* and *tho* mutations (*Collins et al, 2007* – PMID: 17314980; *Wilmes et al, 2008* – PMID: 19061648), alternative NPC tethering strategies would have to be implemented, which would fall beyond the scope of this manuscript.

Figure R1. A hsRNH1-Nup60 fusion does not tether RNase H1 to NPCs in a homogenous manner.

Representative images of cells of the indicated genotypes (*top*, GFP channel; *bottom*, merge with Differential Interference Contrast images). The hsRnh1-GFP-Nup60 fusion is expressed from a centromeric plasmid, under the control of the *NUP60* promoter (*left panel*). *NUP60-GFP* cells express a tagged version of Nup60, driven by its chromosomal promoter (*right panel*). Scale bar, 5µm.

3) The authors should, in general, provide a clearer description on the data presented in the figures, both through the addition of detail in the figures and the figure legends. For example, in Fig. 2D there is no description of the data shown (e.g. what is shown on the y-axis?). In Fig 4A show mass markers. What is the molecular mass of the Rfa1-SUMO? Please generally reexamine the figures and legends.

We have modified figures and legends to improve their clarity. In particular, we have labeled the axes in **Fig. 2d** (Nic96 ChIP-seq coverage for the y-axis; scale in base pairs for the x-axis) and indicated molecular weights in western blot figures (**Fig. 4a**; **Suppl. Fig. 4a-b**). The apparent molecular mass of the Rfa1-SUMO species is ~90kDa, which is consistent with mono-SUMOylation (Rfa1: 70kDa; apparent molecular weight of SUMO: 15-20kDa), as now stated in the figure legend.

4) The conclusion that SUMOylated Rfa1 cannot be detected in WT cells is confounded by reduced levels of Rfa1 observed in WT cells when compared to that in *ulp1-333* cells (see inputs of Fig.4A). This raises the question as to whether Rfa1 is stabilized in *ulp1-333* cells. It would be useful to include loading controls for these input fractions.

We now provide loading controls (Dpm1) for the input fractions of this experiment (**Revised Fig. 4a**). Reduced protein levels are observed for both Rfa1 and the loading control in the *wt* input fraction, indicating that the *ulp1-333* mutant does not actually lead to Rfa1 stabilization. Two other independent biological replicates of the SUMO pull down experiment are now also provided as **Novel Suppl. Fig.**

4a-b, confirming this observation and featuring the detection of enhanced Rfa1 SUMOylation in *ulp1* cells upon HS.

Moreover, other groups have detected Rfa1 SUMOylation in WT cells using SUMO-His pull-downs (see Fig.2F and S1 of Psakhye and Jenetsch. 2012. Cell. 151, 807–820). Establishing the assay for detecting Rfa1-SUMO in WT cells before and after HS would allow the authors to examine the requirement for Mms21 in this event. I would ask the authors to address why they have not pursued this further.

In the referenced study from the Jentsch lab, SUMO pull-downs are performed from cells overexpressing SUMO, a strategy that indeed allows the detection of low levels of Rfa1 SUMOylation in *wt* cells, in the absence of any stress or genotoxic treatment. However, since SUMO overexpression strongly alter natural SUMOylation patterns (see for example Ulrich and Davies, 2009 – PMID: 19107412), we have decided to express tagged SUMO from its endogenous promoter for our pull-down assays, as performed in studies from the Zhao lab (e.g. Cremona et al, 2012 – PMID: 22285753). In this situation, Rfa1 SUMOylation is reportedly undetectable in unchallenged *wt* cells (see Fig. S1A in Cremona et al, 2012), as confirmed by our multiple experiment trials. Of note, although we also manage to detect Rfa1 SUMOylation in *wt* cells upon SUMO overexpression (see below, Fig. R2, lanes 1-2), this experimental setting fails to detect the enhanced modification scored in heat-shocked cells in the presence of natural SUMO levels (Fig. 4a, Suppl. Fig. 4a and Fig. R2, lanes 5-6). For this reason, we have solely investigated Rfa1 SUMOylation in cells expressing endogenous levels of SUMO.

Figure R2 - SUMOylated Rfa1 is only detected in *wt* cells upon SUMO overexpression.

Western blot detection of Rfa1 in input fractions (bottom panel) or purified SUMO-conjugates (top panel) obtained from the indicated strains. Wild-type (*ULP1*) or *ulp1-333* cells expressing His-SMT3 (His-SUMO) either from the *CUP1* strong promoter (*CUP1_{prom}*; Bretes et al, 2014 – PMID: 24500206) or the endogenous *SMT3* promoter (*SMT3_{prom}*, as in this study) were grown at 25°C or heat shocked at 37°C for 15min (heat-shock). The positions of unmodified and mono-SUMOylated Rfa1 are indicated, as well as molecular weights (kDa). Dpm1 serves as a loading control. Lanes 4-6 correspond to the biological replicate #3 of the Rfa1 SUMOylation assays (featured as Suppl. Fig. 4b).

5) There appears to be significantly different percent levels of the *HSP104* locus recovered in the mutants between experiments. See Fig.4 panels D-G. Why is this the case? These experiments do not appear to be very reproducible.

We have reproducibly observed that the increased occurrence of *HSP104* in heavy chromatin fractions scored in *tho* mutants is more pronounced in some genetic backgrounds, with a 5-10-fold enrichment in BY-derivatives (e.g. Fig. 2c, Fig. 3c-d-f, Fig. 4g) vs. a 20-40-fold in W303 derivatives (used to assess the effect of *mms21-11*, *smt3-KR* and *rfa1-KR* mutations in Fig. 4d-f). Importantly, chromatin fractionation was always compared within groups of isogenic strains, in which *tho* alleles were re-introduced for comparison. This is now clarified in the Fig. 4 legend.

6) In Fig.5A, the authors' conclusion that the double mutants *tho rfa1-D228Y* and *tho mms21-11* show growth defects at 25C relative to their single mutant counterparts is not supported by the data. The data at 37C is consistent with their conclusion, with the exception of the *tho slx8* mutant, which shows no difference with the *tho* mutant.

We now also provide images of growth assays performed at 30°C, in which the genetic interaction between *tho* and *rfa1-D228Y*, *mms21-11* or *slx8* mutations is more clearly visualized (Revised Fig. 5a). We have now corrected the text to describe these genetic interactions in an accurate manner (Results section, p12).

7) In Fig. S1F, please shown significance of differences.

An additional replicate is now featured for Suppl. Fig. 1f, together with the according statistical analysis (Mann-Whitney-Wilcoxon test).

8) I would suggest that the authors use a single term, i.e. 'nuclear pore complex or NPC', when referring to nuclear pore complexes and avoid using terms like 'nuclear pore' or 'pore'.

This has been corrected throughout the text.

Reviewer #2

In their manuscript "A R-loop sensing pathway mediates the relocation of transcribed genes to nuclear pore complexes", Arianna Penzo and co-authors correlate R-loop accumulation on chromatin with "gene gating" of the respective loci to nuclear pores. They further demonstrate a dependence on the R-loop-sensing protein RPA, which requires its mono-SUMOylation by Mms21 and interaction with SUMO-interacting domain-containing NPC components. They go on to show that localization to the NPC has a favourable effect at R-loop containing genes, reducing R-loop induced recombination and genotoxicity.

I really like this study – quite a bit of the data is correlative, but it is thoroughly done and nicely controlled; the proposed mechanism explains the observations start to finish and is well supported by the data. The authors do not go so far as to claim it, but I suspect that their proposed mechanism – or aspects of it – could well turn out to be common to other gene gating pathways. It also raises interesting questions about how different "types" of R-loops are differentiated (rDNA is very R-loop prone, but apparently safely tucked away in the nucleolus) and how gene localization impacts genotoxicity, which will certainly be addressed in future studies. The manuscript is well written, experimental conditions are explained with sufficient detail in the Figure legends / M&M section, statistical analyses have been performed throughout, and the figures look polished.

The following comments arise from curiosity rather than being meant as a criticism of the work:

We would like to acknowledge Reviewer #2 for their interest and positive evaluation of our work. We have answered to their comments by providing additional experiments or analyses, as indicated below.

1) If I understand it correctly, presence in the HC fraction is specific to heat-shock genes in the mutants under heat shock, which have exceptionally high levels of NPC association. The negative controls (Suppl Fig 2c) include some housekeeping genes and intergenic genes, but have the authors tested any of their model R-loop rich genes, like the intronless YAT1 construct, or the highest-ranking genes based on Nic96-myc occupancy? Would these also be found in a HC fraction (% at nuclear periphery appears to be in a similar range)? Or are heat-shock genes in the mutants under heat shock a special / extreme case?

We have observed that transcriptional induction of the intronless YAT1 gene in R-loop forming *tho* mutants also leads to its increased occurrence in the HC fraction (**Novel Suppl. Fig. 2g**), albeit to a lesser extent as compared to HS genes. It is likely that HS genes actually represent a rather extreme situation, possibly in line with their exceptional high-frequency promoter firing (*Mouaikel et al, 2013* – PMID: 24210826), as now proposed in the text (**Suppl. Fig 2 legend**).

2) I noticed that the LacO/GFP-LacI signal for the YAT1 construct at the nuclear periphery does not markedly colocalize with the Nup signal (Fig 1g), at least for the two cells shown. This would suggest that the interaction with the NPCs is rather labile while still serving to enrich them in the peripheral zone. There are no example pictures included for the HSP104 locus, but have the authors ever scored the degree of colocalization of GFP-LacI spots with NPCs within the peripheral zone and would this vary with conditions? Just wondering whether this would show a (even) stronger correlation with % in HC, assuming that % in HC reflects genes that are actually bound to an NPC.

The Nup signal can be viewed as a marker for the nuclear periphery (*i.e.* zone 1 for our quantifications) or a proxy of NPC density, since it allows to visualize NPC patches rather than individual NPCs (which are not resolved in our microscopy conditions). The Nup signal intensity is thus highly variable along the nuclear rim, precluding accurate colocalization measurements. We now provide single channel images for Nup49-mCherry and LacO/GFP-LacI, showing that peripheral YAT1 loci actually colocalize with Nup staining, even if not in the most prominent NPC patches (**Novel Suppl. Fig. 1h**).

To more precisely evaluate the colocalization of relocalized loci with NPCs in our imaging assays, we took advantage of the *nup133Δ* nucleoporin mutant, which leads to the formation of discrete NPC clusters at the nuclear periphery, a classical strategy to distinguish NPCs from other components of the NE (*e.g. Nagai et al, 2008* – PMID: 18948542) (**Novel Suppl. Fig. 2e-f**). In this strain, repositioning of HSP104 to the NE largely coincides with its recruitment within NPC clusters, correlating with the results from the fractionation assay.

As a complementary approach to directly score gene-NPC interactions, we also performed Nic96 (NPC) ChIP-qPCR in different mutants (**Novel Suppl. Fig. 2i**). These experiments further confirmed the association of the NPC with several HS-induced loci and its dependence on R-loop accumulation (see also response to Reviewer #1).

3) All the experiments supporting the RPA/SUMO-dependent mechanism were carried out using the HSP104+tho-mutant+heat shock system, which may show the strongest phenotype, but is also slightly artificial. It would seem obvious to also test the effect of at least some mutants along the *rfa1/mms21/slx* axis on a model R-loop rich gene (e.g. the highest-ranking genes based on Nic96-myc occupancy, or the intronless *YAT1* construct). Has this been done?

As pointed above (1), while co-fractionation with NPCs is also observed for the intronless *YAT1* construct, it is less pronounced than for HS genes. Testing for other genes, beyond the HS regulon, in mutants in the RPA/Mms21/Slx axis would thus have required other assays, e.g. microscopy or ChIP, involving additional strain construction that was not compatible with the revision timeframe.

Yet, in support for a role of this pathway beyond the HS situation, we now report that the genetic interaction between the *tho* R-loop accumulating mutant and *rfa1*, *mms21* or *slx8* alleles is also observed in unstressed growth conditions (30°C; **Novel Fig. 5a**).

4) Figure 1b visualizes a rough correlation of Nic96-myc occupancy with levels of R-loops at the locus. Because there appears to be an obvious difference in the pattern of Nic96 across intronless and intron-containing genes – covering the entire transcription unit in the former vs. only the 3' half in the latter (suggesting exclusion from the area that contains the intron) – do any of the R-loop data used for the ranking show a similar pattern (for example, in a metagene plot)? To note, the sites of highest Nic96 signal do not markedly colocalize with the sites of highest RPA signal (compare Figure 1b and Suppl Figure 3b) – for the highest-ranking genes, the pattern even appears to alternate (may be wrong, limited by resolution of Suppl Figure 3b).

Following this remark, we now provide a heatmap analysis of R-loop distribution (**Novel Fig. 1c**), side by side with the Nic96 (NPC) occupancy heatmap (**Fig. 1b**). Comparison of both patterns shows a consistent correlation between R-loop distribution and Nic96 enrichment, with notable low levels of both features in the intron (5' half) of intron-containing loci, further supporting the relationship between hybrid accumulation and NPC association.

As pointed by the reviewer, the correlation with RPA occupancy is less precise, raising the possibility that RPA also associates to genomic regions independently of R-loop formation (as now proposed in the corresponding figure legend).

Minor points:

Abstract: “preventing R-loop-dependent relocation leads to lethality” – This only applies to *tho* mutants, which have an exceptionally high R-loop burden. Please clarify.

This has been corrected as follows: “Preventing R-loop-dependent relocation leads to lethality in hybrid-accumulating conditions” (**Abstract**).

Results, line 212: I believe reference 62 (Reusswig et al. 2020) was inadvertently dropped at the end of the sentence. It clearly belongs there, and is included in this order, but does not appear in the manuscript before the M&M section.

This formatting error has been corrected (now line 220).

Discussion, line 376: Areas of triplet repeat expansion have been shown to be associated with R-loops, at least in some cases (PMID: 24787137). The authors may want to include the reference.

As suggested, we now quote the literature indicating that TNR-containing genes form R-loops (*Groh et al., 2014* – PMID: 24787137; *Su and Freudenreich, 2017* – PMID: 28923949).

Figure 1d and Suppl. 3b: For the “ranked by R-loop levels” score used to sort the genes, the Figure legends simply point to reference 38 (suppl ref 2), Wahba et al. 2016, stating “as previously measured”. While the reference contains the corresponding S1-DRIP-seq data, it does not directly provide a score for R-loop levels for each gene nor any details how such a score would be calculated. I assume this would be the integrated DRIP-seq counts over transcription unit +/- 1000 bp, which are the basis for the heat map sorting in Figure 4A of that reference? Please clarify.

The R-loop values used in the original submission had been calculated as in our previous study (*Bonnet et al, 2017* – PMID: 28757210), based on integrated DRIP-seq counts over hybrid peaks (defined by

peak calling in *Wahba et al, 2016*). We have now integrated DRIP-seq counts over the whole transcription unit in this revised version where we feature the corresponding R-loop heatmap in **Fig. 1c** (as now clarified in the **Methods** section). Note that the gene ranking based on R-loop levels is marginally changed with this modified analysis, leading to slightly updated Nic96 and RPA heatmaps (**Fig. 1b** and **Suppl. Fig. 3a**), with no impact on the conclusions.

Figure 2d: Since all the other panels are nicely labelled with the experiment type, please also add “Nic96 occupancy” to panel d.

This has been corrected.

All Figures labelled “DNA in HC fraction” (Figures 2c, 3b,c,d,f, 4d,e,f,g, ... & Supplements): “% relative to wt” does not make any sense – after normalization, it’s just a fold-change in the relative HC fraction association. Please relabel. It’s not quite clear to me why one would want to normalize this to wt in the first place? Just plotting the % HC / total would facilitate comparison to the peripheral localization data.

We have corrected these figure legends as follows: “*HSP104 in P14K/total, relative to wt*”. We believe that the normalization to the *wt* allows for a better comparison in between fold-enrichments in HC fractions for different experiments. Of note, although the outcome of both fractionation and microscopy assays are qualitatively similar, they do not compare strictly at the quantitative level, notably because the range of detection of changes in localization in the imaging analysis is narrower (as pointed in the **Results section**, lines 116-117).

Figure 5c and Suppl Fig 1g: “Recombination frequency” is missing a “b”.

This has been corrected.

Reviewer #3

In the manuscript by Penzo et al, there is exciting data showing that the formation of R-loops results in the repositioning of the R-loop harboring locus to the nuclear pore. This is elegantly demonstrated using a host of techniques ranging from ChIP-seq to microscopy to genetic manipulation. Using ChIP with the Nic96 protein it is shown that R-loop harboring (largely intronless) genes are found associated with the pore, this is also consistent with other nuclear pore complex members. The authors then show that the Yat1 locus localizes to the nuclear periphery upon transcription induction however it fails to do so when an intron is inserted into the locus, which prevents R-loop accumulation. R-ChIP is employed to demonstrate the R-loops accumulate at heat shock loci and that these loci accumulate in the heavy chromatin fraction of cell extracts (consistent with pore association). Importantly, this effect is enhanced when the THO complex is mutated and R-loops accumulate to even higher levels. The relocalization to the pore is likely independent of DNA damage as it is also seen in G1 arrested cells, as well as in *rad2*, *mlh2* and *fcy1* mutants. Interestingly, the presence of RPA is also required for the nuclear repositioning of R-loops. Indeed, in *rfa1-D228Y* mutants the repositioning of the HSP104 locus is faulty following heat shock in a *tho* mutant background. The SUMOylation of Rfa1 seems to be critical for the nuclear pore localization of R-loops as an *Rfa1-4K* mutant as well as *mms21* mutations prevent it. Finally it is suggested in the final figure that re-localization of R-loops to nuclear pores aids in promoting genome stability.

Altogether this is a manuscript of very high quality and it is very well written. All experiments are well controlled and the message is clear. I feel that the functional implications of R-loop localization are still poorly understood, although Figure 5 attempts to address this, it is not completely convincing. Nonetheless I feel that this manuscript should be directly accepted for publication and will be interesting for a very general audience. I would have one minor suggestion that could be attempted, however it is not essential.

In Figure 1, it would have been nice to show that RNase H1 overexpression reduces the pore association of certain R-loop containing genes. Again, I understand the caveats associated with such an experiment.

Congratulations on a beautiful manuscript.

We acknowledge the Reviewer for their enthusiastic evaluation of our study.

In line with a similar remark from Reviewer #1, we have now performed Nic96 (NPC) ChIP-qPCR in conditions of HsRNH1 overexpression (featured as **Novel Suppl. Fig. 2i**). This experiment revealed decreased Nic96 ChIP levels on HS genes upon R-loop removal, confirming that their association with the NPC is R-loop dependent (see also answer to Reviewer #1, point 1.).

REVIEWERS' COMMENTS

Reviewer #1 (Remarks to the Author):

The authors' revised manuscript is significantly improved. I recommend publication.

Rick Wozniak

Reviewer #2 (Remarks to the Author):

It was a good manuscript before the review, and it's been strengthened further by the additional data. I would like to thank the authors for going the extra mile to satisfy my curiosity - they have addressed all the points I have raised.

Reviewer #1 (Remarks to the Author)

The authors' revised manuscript is significantly improved. I recommend publication.

Rick Wozniak

Reviewer #2 (Remarks to the Author)

It was a good manuscript before the review, and it's been strengthened further by the additional data. I would like to thank the authors for going the extra mile to satisfy my curiosity - they have addressed all the points I have raised.

We acknowledge both reviewers for their positive assessment of our manuscript.